# Impacts of the photo-driven post-depositional processing on snow nitrate and its isotopes at Summit, Greenland: a model-based study

Zhuang Jiang[1], Becky Alexander[2], Joel Savarino[3], Joseph Erbland[3], Lei Geng[1,4,5,6*]

[1]Anhui Province Key Laboratory of Polar Environment and Global Change, School of Earth and Space Sciences, University of Science and Technology of China, Hefei, Anhui, China

[2]Department of Atmospheric Sciences, University of Washington, Seattle WA, USA

[3]Univ. Grenoble Alpes, CNRS, IRD, G-INP, Institut des Géosciences de l'Environnement, Grenoble, France

[4]Laboratory for Ocean Dynamics and Climate, Pilot Qingdao National Laboratory for Marine Science and Technology, Qingdao, Shandong, China

[5]CAS Center for Excellence in Comparative Planetology, University of Science and Technology of China, Hefei, Anhui, China

[6]Hefei National Laboratory for Physical Sciences at the Microscale, University of Science and Technology of China, Hefei, Anhui, China

*Correspondence to*: Lei Geng (genglei@ustc.edu.cn)

**Abstract.** Atmospheric information embedded in ice-core nitrate is disturbed by post-depositional processing. Here we used a layered snow photochemical column model to explicitly investigate the effects of post-depositional processing on snow nitrate and its isotopes ($\delta^{15}N$ and $\Delta^{17}O$) at Summit, Greenland where post-depositional processing was thought to be minimal due to the high snow accumulation rate. We found significant redistribution of nitrate in the upper snowpack through photolysis
and up to 21 % of nitrate was lost and/or redistributed after deposition. The model indicates post-depositional processing can reproduce much of the observed $\delta^{15}N$ seasonality, while seasonal variations in $\delta^{15}N$ of primary nitrate is needed to reconcile the timing of the lowest seasonal $\delta^{15}N$. In contrast, post-depositional processing can only induce less than 2.1 ‰ seasonal $\Delta^{17}O$ change, much smaller than the observation (9 ‰) that is ultimately determined by seasonal differences in nitrate formation pathway. Despite significant redistribution of snow nitrate in the photic zone and the associated effects on $\delta^{15}N$ seasonality,
the net annual effect of post-depositional processing is relatively small, suggesting preservation of atmospheric signals at the annual scale under the present Summit conditions. But at longer timescales when large changes in snow accumulation rate occurs this post-depositional processing could become a major driver of the $\delta^{15}N$ variability in ice core nitrate.

## 1. Introduction

Nitrate ($NO_3^-$) is one of the most abundant and commonly measured species in ice cores. One of the major subjects of ice-core nitrate studies involves its oxygen isotope mass-independent fractionation signal ($\Delta^{17}O = \delta^{18}O - 0.52 \times \delta^{17}O$), which is a proxy of atmospheric oxidation capacity (Alexander & Mickley, 2015; Alexander et al., 2004; Geng et al., 2017). Ice-core $\delta^{15}N(NO_3^-)$ records have also been studied but the interpretation remains immature and sometimes conflicting (Freyer et al., 1996; Geng et al., 2014, 2015; Hasting et al., 2005, 2009). There are many factors, e.g., $NO_x$ sources, atmospheric chemistry and transport, deposition and post-depositional processing of nitrate, affecting ice-core nitrate and its isotopes (Geng et al., 2014, 2015; Hastings et al., 2004, 2005; Morin et al., 2008; Wolff et al., 2008).

Deposition of atmospheric nitrate to snow is not irreversible. The ultimate source of snow nitrate in the polar regions is from tropospheric long-range transport and stratospheric denitrification (Goto-Azuma and Koerner, 2001; Legrand and Delmas, 1986), which can be termed as primary nitrate ($F_{pri}$) (Erbland et al., 2013). After deposition, nitrate undergoes post-depositional processing which causes changes in its concentration and isotopes (Blunier et al., 2005; Erbland et al., 2013; Frey et al., 2009). Post-depositional processing of snow nitrate includes physical release (i.e., desorption and evaporation) and ultraviolet photolysis. Both processes result in loss of snow nitrate and isotope fractionations of nitrogen and oxygen. However, laboratory experiments and model calculations indicate a minor influence of the physical processes, with photolysis dominating post-depositional processing (Berhanu et al., 2014; Erbland et al., 2013; Frey et al., 2009; Zatko et al., 2016).

Snow nitrate photolysis occurs when it is exposed to sunlight at wavelengths less than 345 nm (Chu & Anastasio, 2003). The dominant photolysis product is $NO_2$, which is effectively transported to the overlying atmosphere via diffusion or wind pumping (Zatko et al., 2013) and impacts local atmospheric oxidation environment (Thomas et al., 2012). The released $NO_2$ can reform $HNO_3$ (i.e., the snow sourced nitrate hereafter) in the overlaying atmosphere, which is then redeposited to or exported from the site of photolysis. The above-mentioned processes form a cycle of nitrate between the air-snow interface, resulting in redistribution of nitrate in snowpack.

The photolysis also causes isotope fractionation. The isotope fractionation factors ($\varepsilon_p$) associated with snow nitrate photolysis are ($-47.9 \pm 6.8$ ‰) and $-34$ ‰ for $\delta^{15}N$ and $\delta^{18}O$, respectively, for local conditions at Dome C, East Antarctica (Berhanu et al., 2014; Blunier et al., 2005; Frey et al., 2009). These large negative values indicate the photolysis would enrich nitrate remaining in snow with heavier isotopes (i.e., $^{15}N$ and $^{18}O$). In comparison, $\Delta^{17}O(NO_3^-)$ in snow will not be directly disturbed by photolysis. However, part of the photo-product can undergo recombination reactions within snow grains to reform nitrate (i.e., the cage effect) (McCabe et al., 2005; Meusinger et al., 2014). This process results in exchanges of oxygen atoms with snow and decreasing $\Delta^{17}O(NO_3^-)$ and $\delta^{18}O(NO_3^-)$. In addition, once the photo-product $NO_2$ is released to the overlying atmosphere, it is rapidly converted to nitrate and carries different $\Delta^{17}O(NO_3^-)$ values from its precursors. These isotope effects have been documented in multiple snowpack studies on the East Antarctic Plateau, with increasing $\delta^{15}N$ and decreasing $\Delta^{17}O(NO_3^-)/\delta^{18}O(NO_3^-)$ with depth (Erbland et al., 2013; Frey et al., 2009; Shi et al., 2015).

The degree of post-depositional processing and the induced effects on snow nitrate and isotopes vary site by site, depending on several factors including actinic flux, snow properties (e.g., density, light-absorbing impurities, specific surface

area) and snow accumulation rate (Zatko et al., 2013). Actinic flux describes the light intensity reaching snow surface, while snow properties determine the penetration of light in snow. Actinic flux decreases exponentially from the snow surface, and the depth of the snow photic zone is defined as 3 times the e-folding depth of the actinic flux (Erbland et al., 2013). Snow accumulation rate determines the residence time of nitrate in the photic zone where photolysis occurs, and thus at sites with high snow accumulation rate the degree of post-depositional processing will be limited (Erbland et al., 2013; Noro et al., 2018; Shi et al., 2015).

Summit, Greenland is a typical high snow accumulation site (250 kg m$^{-2}$ a$^{-1}$, Dibb et al., 2004), where snowpack and ice core nitrate isotopes records have been studies but the interpretation on $\delta^{15}$N remains conflicting (Geng et al., 2014, 2015; Hastings et al., 2005,2009). Hastings et al. (2005) proposed the glacial-interglacial $\delta^{15}$N difference observed in the GISP2 ice core was due to changes in NOx source strengths despite the fact that there is no known NOx source carrying high enough $\delta^{15}$N to explain the glacial $\delta^{15}$N value ((28.4 ± 1.1) ‰). In contrast, Geng et al. (2015) concluded that changes in the degree of post-depositional processing between the glacial and interglacial climates can explain the more enriched $\delta^{15}$N in the glacial period. On seasonal time scales, there are also distinct variations in nitrate $\delta^{15}$N and $\Delta^{17}$O at Summit (Geng et al., 2014; Hastings et al., 2004; Jarvis et al., 2009; Kunasek et al., 2008). The seasonality of $\delta^{15}$N was originally attributed to variations in NO$_x$ sources (Hastings et al.,2004) and the $\Delta^{17}$O was suggested to be mainly caused by changes in atmospheric nitrate formation pathways (Kunasek et al., 2008). However, the effect of post-depositional processing on seasonal $\delta^{15}$N variation was not thoughtfully examined by Hastings et al. (2004). Two later studies by Fibiger et al. (2013, 2016) examined nitrate isotopes in the atmosphere and surface snow (< 3 cm depth) at Summit to investigate the effects of post-depositional processing, and concluded that the effects of post-depositional processing on nitrate isotope preservation is negligible. These studies, however, relied only on surface snow samples and didn't cover a full year of nitrate deposition. The snow photic zone at Summit is 30 to 40 cm deep (Galbavy et al., 2007), and thus surface snow samples cannot readily reflect the full degree of post-depositonal processing as the effects increase with time in the snow photic zone and thus depth. In particular, Fibiger et al. (2013) used the observed linear relationship between surface snow $\Delta^{17}$O(NO$_3^-$) and $\delta^{18}$O(NO$_3^-$) to exclude the effects of post-depositional processing on nitrate isotopes at Summit, Greenland. This approach is flawed as neither $\delta^{18}$O(NO$_3^-$) nor $\Delta^{17}$O(NO$_3^-$) is a good indicator of post-depositional processing because the oxygen isotopes are mainly controlled by atmospheric processes (Alexander et al., 2020; Kunasek et al., 2008). This is why a strong linear relationship between $\Delta^{17}$O(NO$_3^-$) and $\delta^{18}$O(NO$_3^-$) is observed in atmospheric and surface snow samples at Dome C, Antarctica where severe post-depositional processing of nitrate occurs (Erbland et al., 2013; Frey et al., 2009). At sites with extremely low snow accumulation rates (e.g., Dome C and Dome A in Antarctica) the cage effect would cause apparent changes in $\delta^{18}$O(NO$_3^-$) and $\Delta^{17}$O(NO$_3^-$) in samples at depth but not at the surface (Erbland et al., 2013; Shi et al., 2015).

Given the distinct seasonal differences in actinic flux in the polar regions including Summit, Greenland, seasonal differences in the degree of post-depositional processing and its effects on snow nitrate isotopes should be examined. In fact, observations at Summit indicate that $\delta^{15}$N in surface snow nitrate is negative during most of the year with an annual mean of (−6.2 ± 1.1) ‰ (Jarvis et al., 2009), while in bulk snowpack (0 to 1 m) the annual mean $\delta^{15}$N is (0 ± 2.4) ‰ (Jarvis et al., 2009)

and $(0 \pm 6.3)$ ‰ (Geng et al., 2014). During spring and summer when snow photochemistry is most active, $\delta^{15}N$ in surface snow is $(-5.8 \pm 0.7)$ ‰, while in snowpack (average of two springs at depths of 0.15 and 0.9 m, respectively) is $(5.6 \pm 1.8)$ ‰. These differences between nitrate at the surface and at depth suggest enrichment in nitrate $\delta^{15}N$ after deposition and recycling

at the surface, and are consistent with the known isotope effects of post-depositional processing. In addition, Burkhart et al. (2004) and Dibb et al. (2007) have observed <7 % to 25 % loss of nitrate after deposition at Summit. This is close to the estimate of 16-23 % loss based on ice-core $\delta^{15}N(NO_3^-)$ (Geng et al., 2015). These results are also qualitatively consistent with the observations of $NO_2$ and HONO fluxes from snowpack at Summit which were attributed to snow nitrate photolysis (Dibb et al., 2002; Honrath et al., 2002). The spatial variations in the photo-driven nitrate recycling at the air-snow interface and its

impact on snow $\delta^{15}N(NO_3^-)$ in Greenland have been studied by Zatko et al. (2016) using a global 3-D chemical transport model (GEOS-Chem). Their model captures the increasing trend in snow $\delta^{15}N(NO_3^-)$ from costal to inland as snow accumulation rate decrease that enhances the degree of post-depositional processing. This is consistent with field studies of snow nitrate $\delta^{15}N(NO_3^-)$ in West Greenland (Curtis et al., 2018). But the model treated snowpack as a whole and didn't specify the behaviours of nitrate at different depths in the photic zone, and cannot distinguish seasonal differences. In addition, it did not

incorporate isotope fractionation associated with photolysis, but instead using a fixed fractionation constant and a Rayleigh fractionation model to calculate the changes in isotope with mass loss.

In order to investigate the impacts of snow nitrate photolysis on the preservation of nitrate and its isotopes on the seasonal time scale at Summit, Greenland, we used a snow photochemical column model to simulate the recycling of nitrate at the air-snow interface. We use the model to quantify to what degree the magnitude of the observed seasonality of nitrate isotopes at

Summit can be explained by post-depositional processing. The model was built to explicitly investigate the loss of snow nitrate due to photolysis and quantify the induced isotope effects with layer specific calculations (i.e., changes in $\delta^{15}N$ and $\Delta^{17}O$ after deposition). Comparison of the model results with observations should add insight into the preservation of nitrate at high snow accumulation sites and shed light on the interpretation of ice-core nitrate and its isotopes.

## 2. Model description

TRANSITS (Transfer of Atmospheric Nitrate Stable Isotopes To the Snow) is a multi-layer, 1-D model that simulates nitrate recycling at the air-snow interface, and its preservation in snow including its isotopes (Erbland et al., 2015). The model divides a year into 52-time steps (i.e., weekly resolution) and at each step the snowpack is divided into 1 mm layers where photolysis of nitrate is calculated according to the depth-dependent actinic flux and nitrate concentration. The produced $NO_2$ is transported to the overlying atmosphere where it is re-oxidized to nitrate. At the next time step, a portion of the reformed

nitrate together with primary nitrate originating from long-range transport deposit to snow surface. When snowfall occurs, the snowpack moves down and the newly deposited snow is immediately re-divided into 1 mm layers. Nitrate is considered as archived once it is buried below the photic zone.

At each step, the model also calculates the isotope effects. In the model, nitrogen isotope fractionation mainly occurs during the photolysis with a wavelength sensitive fractionation constant $\varepsilon_p$, and another fractionation occurs during nitrate

deposition with a fractionation constant $\varepsilon_d$. The oxygen isotope effect is only calculated for $\Delta^{17}O$, which is caused by 1) exchange of oxygen atoms with water during the photolysis (i.e., the cage effect), and 2) local atmospheric NO-NO$_2$ cycling and the subsequent conversion of NO$_2$ to HNO$_3$.

To run the model, actinic flux and its e-folding depth in snowpack, snow accumulation rate, as well as other atmospheric properties including the boundary layer height, surface ozone and HO$_x$ concentrations are needed. Additional model inputs are

the flux of primary nitrate from long-range transport and its isotopic composition (i.e., $\delta^{15}N$ and $\Delta^{17}O$). In this study we focus on the seasonal changes in isotopes caused by post-depositional processing, making the results independent of $\delta^{15}N$ and $\Delta^{17}O$ of primary nitrate.

In this study, we run the model from the year 2004 to 2007 constrained by local observations at Summit. The modeled snow nitrate concentration and isotope profiles were compared with observations in Geng et al. (2014).

**2.1 Model inputs**

**2.1.1 Atmospheric characterizations**

The overlying atmosphere at Summit was assumed to be a one-dimensional box with constant boundary layer height of 156 m (Cohen et al., 2007), where primary and the snow-sourced nitrate are assumed to be well mixed. Weekly air temperature, pressure, surface ozone concentration and total column ozone (TCO) at Summit were obtained from the NOAA ozonesonde

dataset (https://www.esrl.noaa.gov/gmd/ozwv/ozsondes/sum.html). Concentrations of local atmospheric oxidants including O$_3$, OH, peroxyl radicals and BrO are needed to calculate the cycling of NO-NO$_2$ and the conversion of NO$_2$ to HNO$_3$. At Summit, there are no long-term observations of OH and peroxyl radicals (RO$_2$, HO$_2$) which are necessary to calculate the atmospheric transformation of NO$_x$ to HNO$_3$, so we estimated their mixing ratio by assuming a linear relationship with local $J_{(NO2)}$. More specifically, the photolysis rate constant of NO$_2$ were first calculated using local actinic flux, and the concentrations of OH

and peroxyl radicals were calculated by assuming their linear relationships with $J_{(NO2)}$ (Kukui et al., 2014), respectively. Diurnal observations of OH and peroxyl radicals exist at Summit with noon values of $6.3 \times 10^6$ and $2.4 \times 10^8$ molecule cm$^{-3}$ (Sjostedt et al., 2007), respectively. We used these values to justify the calculated OH and peroxyl radical values by applying scaling factors to match them with the observations. We set a constant BrO concentration of 2 pptv in summer and zero in other seasons, given the observed summer BrO concentration (1-3 pptv) at Summit (Fibiger et al., 2016).

The mass balance of nitrate in snowpack and the overlying atmosphere is controlled by nitrate flux in and out of the snow. We denote the nitrate fluxes as ''FY'' following Erbland et al. (2015), with $F_{pri}$, FP, FD and FA representing the primary nitrate flux from long range transport, the nitrate flux that originates from photolysis of snow nitrate, the deposition flux of atmospheric nitrate, and the archived snow nitrate flux that is buried under the photic zone, respectively. These fluxes determine the variations of snow and atmospheric nitrate and their isotope compositions.

## 2.1.2 Radiative transfer and nitrate photolysis rate in snow

Downward/upward actinic flux spectrum at the snow surface was calculated using the Troposphere Ultraviolet and Visible (TUV) radiation model (Madronich et al., 1998) constrained by TCO. Radiative transfer inside the snowpack was then computed using the Two-stream Analytical Radiative TransfEr in Snow (TARTES) model (Libois et al., 2013). The attenuation of light in snow is characterized by its e-folding depth, which represents the depth where radiation decreases to 1/e of the surface intensity. Snow e-folding depth depends on its optical properties (e.g., bulk density, snow grain size) and on the concentrations of light-absorbing impurities (Zatko et al., 2013). In this study, for simplification, we set constant snowpack concentrations of the three main snow light-absorbing impurities, soot, dust and organic humic-like substance (HULIS) as 1.4, 138 and 31 ng g$^{-1}$, respectively (Zatko et al.,2013; Carmagnola et al., 2013). Snow density and grain size also impact the e-folding depth. The snow radiation equivalent mean grain radius ($r_e$) is linked to the specific surface area (SSA) of snow grains by $r_e$ =3 / (SSA × $\rho_{ice}$). Since direct observations of SSA of the reported snowpack in Geng et al. (2004) are lacking and only density profile data exists, we used the regression relationship between SSA and $\rho_{snow}$ (SSA = −174.13 × ln($\rho_{snow}$) + 306.4, in unit of cm$^2$ g$^{-1}$ for SSA and g cm$^{-3}$ for density, respectively) from Domine et al. (2007) to calculate SSA. Using the observed snow density, fixed light-absorbing impurity concentrations and the calculated SSA profile, we obtained an e-folding depth of 12.3 cm (at a wavelength of 305 nm, which is the peak wavelength of nitrate photolysis) that is similar to the measured average summer midday value (11.6 cm) at Summit (Galbavy et al., 2007), but lower than the modelled result (15-17 cm) by Zatko et al. (2013). Note Zatko et al. (2013) applied the measured snow $r_e$ profile at Dome C to Summit condition with SSA ranged from 7 to 38 m$^2$ kg$^{-1}$, which was lower than our calculated SSA of 44 to 51 m$^2$ kg$^{-1}$. This likely explains why our calculated e-folding depth was smaller than Zatko et al. (2013) despite using the same impurity content. We also note that the regression relationship between SSA and $\rho_{snow}$ from Domine et al. (2007) was for fresh snow, which may not be suitable for SSA prediction for the whole snowpack. However, using this equation yielded an e-folding depth that is similar to the observations by Galbavy et al. (2007), despite the yielded SSA appears to be larger than the observed values (20 to 40 m$^2$ kg$^{-1}$) by Carmagnola et al. (2013) for a Summit snowpack which has a much lower snow density (averaged 330 kg m$^{-3}$ in the top 50 cm) than ours (averaged 395 kg m$^{-3}$). Nevertheless, given the uncertainties related to the calculation of snow radiative transfer that are currently not well constrained, the regression relationship between SSA and $\rho_{snow}$ used here yielded a reasonable e-folding depth similar to the observations. Improvements can be made if snow physicochemical properties (e.g., SSA, density, and impurities concentrations) can be precisely well constrained by future observations.

The photolysis rate constant of snow NO$_3^-$ was calculated by:

$$J(z) = \int_{280\,nm}^{350\,nm} \Phi(\lambda) \times \sigma_{NO_3^-}(\lambda) \times I(z, \lambda)\, d\lambda \tag{1}$$

Where $I$ is actinic flux, $\Phi$ and $\sigma$ are the quantum yield and absorption cross section of nitrate photolysis, respectively. The absorption cross sections of $^{14}NO_3^-$ and $^{15}NO_3^-$ were from Berhanu et al. (2014). In this study, we used the measured surface snow nitrate photolysis rate constant $j_0(NO_3^-)$ (Galbavy et al., 2007) to constrain the quantum yield at Summit. Galbavy et al. (2007) reported that $j_0(NO_3^-)$ in surface snow at summer noon generally falls in the range of $(1-2) \times 10^{-7}$ s$^{-1}$ with a mean value of $1.1 \times 10^{-7}$ s$^{-1}$. This value corresponds to a quantum yield of 0.002 given typical Summit summer column ozone density (350 DU) and noon solar zenith angle (50 degree). We adopted this value of quantum yield in our model, and according to summer actinic flux, its penetration in snowpack and snow nitrate concentration at Summit, we calculated a summer mean NO$_x$ flux from the snowpack of $(2.96 \pm 0.3) \times 10^{12}$ molecules m$^{-2}$ s$^{-1}$ that is close to the observation of $2.52 \times 10^{12}$ molecules m$^{-2}$ s$^{-1}$ by Honrath et al. (2002).

### 2.1.3 Flux of primary nitrate ($F_{pri}$) and the export fraction

Primary nitrate from long range transport was assumed to be the only external nitrate source for Summit. Given the mean snow accumulation rate (250 kg m$^{-2}$ a$^{-1}$), and the mean snowpack nitrate concentration (117 ng g$^{-1}$) at Summit, a minimum annual $F_{pri}$ of $6.6 \times 10^{-6}$ kgN m$^2$ a$^{-1}$ was estimated and used in the model. This value is at the same order of magnitude ($\approx 2 \times 10^{-6}$ kgN m$^2$ a$^{-1}$) as modeled by Zatko et al. (2016). The seasonal variability of $F_{pri}$ was adjusted to $1.6 \times 10^{-6}$, $2.1 \times 10^{-6}$, $1.6 \times 10^{-6}$ and $1.2 \times 10^{-6}$ kgN m$^{-2}$ season$^{-1}$ for spring, summer, autumn and winter, respectively according to back-trajectory analyses and a regional emission inventory (Iizuka et al., 2018). The values and seasonal variations of $\delta^{15}N$ and $\Delta^{17}O$ of $F_{pri}$ are currently unknown. We set $\delta^{15}N$ and $\Delta^{17}O$ of $F_{pri}$ as 0 and 30 ‰ (close to their average values in snowpack), respectively, throughout the year. This takes the advantage of the model to explicitly assess the effects of the photolysis while excluding other influencing factors. In addition, previous studies proposed $\delta^{15}N$ of snow nitrate at Summit should reflect $\delta^{15}N$ of NO$_x$ sources (Hasting el al., 2004, 2005), so that in order to investigate the sensitivity of snowpack $\delta^{15}N(NO_3^-)$ to $\delta^{15}N$ of $F_{pri}$, we also used the measured $\delta^{15}N$ in surface snow nitrate at Summit that varies seasonally (Jarvis et al., 2009) as a first order estimation of $\delta^{15}N$ of $F_{pri}$. Note this may underestimate $\delta^{15}N$ of $F_{pri}$, as surface snow nitrate could be influenced by snow-sourced nitrate that is in general depleted in $\delta^{15}N$. Nevertheless, we note that $\delta^{15}N$ and $\Delta^{17}O$ of $F_{pri}$ are just starting points to run the model, and the predicted changes caused by post-depositional processing are independent of these values.

Another parameter influencing the preservation of nitrate is the export fraction, $f_{exp}$, which represents the fraction of the snow sourced NO$_x$ and nitrate transported away from the site of photolysis. At the site of photolysis, part of the reformed nitrate in the atmosphere will be exported and which represents the net loss of nitrate through the post-depositional processing. We estimated the export fraction ($f_{exp}$) following the method used by Erbland et al. (2015):

$$f_{exp} = \frac{\frac{1}{\tau_2}}{\frac{1}{\tau_1} + \frac{1}{\tau_2}} \times \left( 1 + \frac{\frac{1}{\tau_1}}{\frac{1}{\tau_3} + \frac{1}{\tau_1}} \right) \tag{2}$$

Where $\tau_1$, $\tau_2$ and $\tau_3$ denote the lifetimes of horizontal transport, oxidation of NO$_2$ by OH radicals and vertical deposition, respectively. $\tau_1$, $\tau_2$ and $\tau_3$ were calculated as follows:

$$\tau_1 = \frac{L}{v_h} \tag{3}$$

$$\tau_2 = \frac{1}{k[OH]} \tag{4}$$

$$\tau_3 = \frac{H}{v_d} \tag{5}$$

Where $H$ and $L$ represent the vertical and horizontal characteristic dimensions of 156 m (average summer boundary layer height at Summit) and 350 km (characteristic length of summit of the Greenland ice cap, Honrath et al., 2002), respectively. $v_h$ is the mean horizontal wind speed at Summit (5 m s$^{-1}$) and $v_d$ is the dry deposition velocity of HNO$_3$ (0.63 cm s$^{-1}$) (Björkman et al., 2013). $k$ is the kinetic rate constant as a function of temperature and pressure for NO$_2$+OH->HNO$_3$ ($3 \times 10^{-12}$ cm$^3$ molecule$^{-1}$ s$^{-1}$ on average in summer, Atkinson et al., 2004). From Eq (2) we obtained a value of 0.35 for $f_{exp}$ in summer conditions and kept it constant in the model simulations. Note this value is irrelevant in winter when photolysis stops, therefore there is no need to consider the seasonal difference of $f_{exp}$. In addition, we note the $f_{exp}$ calculated from the above equations is just a rough estimate as it may oversimplify the processes governing nitrate deposition and chemical loss pathways of NO$_x$. The sensitivity of model results to $f_{exp}$ is discussed in section 3.3.

## 2.2 Calculation of the isotope effects

The nitrogen isotope fractionation constant ($^{15}\varepsilon_p$) during photolysis was calculated from the ratio of $^{14}$NO$_3^-$ and $^{15}$NO$_3^-$ photolysis rates in each snow layer ($^{15}\varepsilon_p = J^{15}/J^{14} - 1$). The deposition of atmosphere nitrate can induce isotope fractionation ($\varepsilon_d$) in $\delta^{15}$N based on simultaneous measurements of atmospheric and surface $\delta^{15}$N(NO$_3^-$) (Erbland et al., 2013; Fibiger et al., 2016). Fibiger et al. (2016) suggested that at Summit the fresh snow NO$_3^-$ is enriched in $\delta^{15}$N by +13‰ compared to atmospheric NO$_3^-$, similar to the observation at Dome C, Antarctica (+10 ‰, Erbland et al., 2013). In contrast, Jarvis et al. (2009) found no difference in $\delta^{15}$N of gas-phase HNO$_3$ and surface snow NO$_3^-$ at Summit. Here we followed Erbland et al. (2013) to set $\varepsilon_d$ as +10‰. We did not use the results from Fibiger et al (2013) which was conducted in spring when photolysis of snow nitrate had already started and disturbed the connection between atmospheric nitrate and that in surface snow. For oxygen isotopes, the $\Delta^{17}$O of the reformed nitrate in the air was assumed to be 2/3 of $\Delta^{17}$O(NO$_2$), which assumes that NO$_2$ + OH is the dominant nitrate production mechanism under sunlight. $\Delta^{17}$O(NO$_2$) was estimated according to the relative importance of O$_3$ and BrO versus HO$_2$ and RO$_2$ oxidation of NO to NO$_2$. The $\Delta^{17}$O value of bulk O$_3$ is taken at 26 ‰ (Vicars & Savarino, 2014), that of BrO is 39 ‰, and other oxidants are 0 ‰. We assumed a cage effect of 15 % following Erbland et al. (2015).

## 2.3 Model initiation

The model was initiated by deposition of primary nitrate mixed with snow-sourced nitrate. A real snowpack with a depth of 2.1 m and known nitrate concentration and isotope profiles (Geng et al., 2014) was set at time ($t$) = 0. Weekly snow accumulation rate was obtained by averaging the observed snow accumulation of the same week (week 1[th] to week 52[th]) of a year over 2003 to 2007 at Summit. Average instead of real accumulation data were used to avoid negative values in some weeks due to wind blowing which causes net loss instead of gain of snow. After a three years simulation, the snow nitrate concentration and isotope profiles above the pre-existing snowpack were sampled from the model to compare with the observations from Geng et al 2014. All parameters need to run the model were listed in Table S1.

## 3 Results and discussion

### 3.1 The simulated snowpack nitrate depth profiles at Summit, Greenland

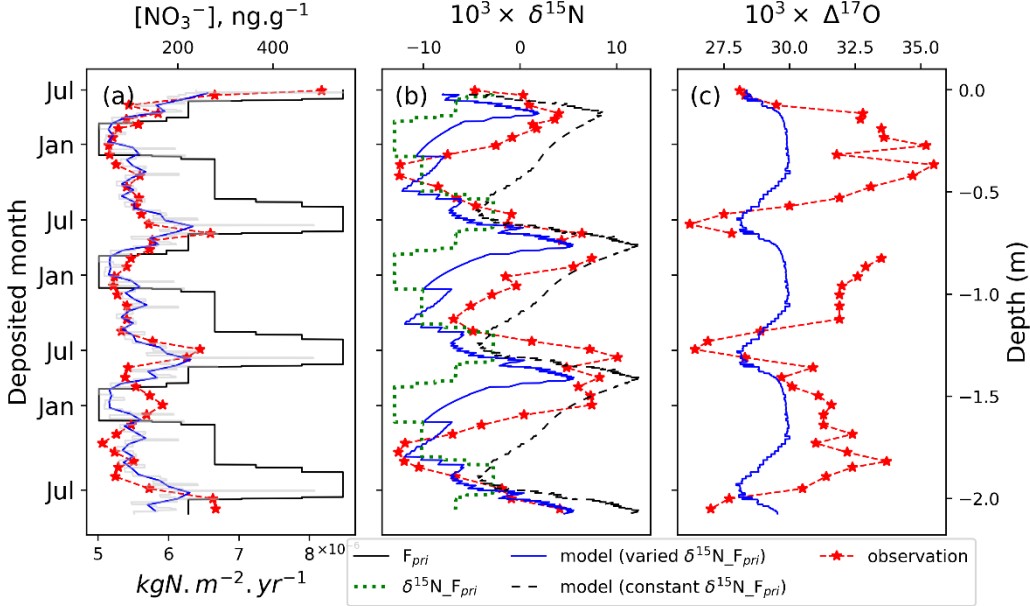

**Figure 1.** Snowpack nitrate concentration and isotopes profiles at Summit, Greenland (red: observations, blue: modeled). The gray curve in (a) is the modeled weekly data while the blue is the monthly average. The green dashed line in (b) represents measured $\delta^{15}N$ in surface snow throughout a year (Jarvis et al., 2009). The measured minimum $\Delta^{17}O(NO_3^-)$ was used as the indicator of June-July when local photochemistry is the most active.

The observed and modeled snowpack nitrate concentration and its isotopes (i.e., $\delta^{15}N$ and $\Delta^{17}O$) from July 2004 to 2007 are plotted in Figure 1. The observations were from a snowpit collected in July 2007 so that the top of the observed profiles

represents a summer, and we used the observed $\Delta^{17}O$ minimum and concentration maximum to identify other summers to match the modeled profiles with the observations. In addition, the depth of the modeled snowpack was adjusted according to the difference in fresh snow density and the measured snow density profile in the upper 2 meters at Summit (Geng et al., 2014).

As shown in Figure 1, nitrate concentrations and isotopes in the modeled snowpack in general display similar seasonal patterns to the observations, except for $\Delta^{17}O$ whose magnitude of seasonal change is much smaller than the observations. The modeled average $NO_3^-$ concentration was (115 ± 65) ng g$^{-1}$, similar to the observation of (117 ± 62) ng g$^{-1}$. The modeled concentration profile displays high variability which is mainly caused by variations in weekly snow accumulation. The modeled results indicate clear summer peaks and winter valleys similar to the observations. In addition, we found with or without seasonal variations in $F_{pri}$, the modeled concentration and isotope profiles were almost identical.

The modeled $\Delta^{17}O(NO_3^-)$ deviated by about 2.1 ‰ from primary nitrate ($\Delta^{17}O(NO_3^-)$ = 30 ‰) in summer. This is consistent with expectations as post-depositional processing will not cause mass-independent fractionation so that it has no direct effects on $\Delta^{17}O$. The model deviation is mainly caused by the reformation of nitrate in the local atmosphere which leads to nitrate with different $\Delta^{17}O$ from primary nitrate. In summer, nitrate reformed in the overlying atmosphere occurs mainly through OH oxidation of $NO_2$. In the model, nitrate formed through this process possessed $\Delta^{17}O$ of (19.6 ± 0.3) ‰ on average. This value is close to the modeled results (18.9 ‰) for summer at Summit by Kunasek et al. (2008) who used a box model and assumed local $NO_x$ chemistry is the only nitrate source. $\Delta^{17}O$ of nitrate formed from local chemistry is lower than that in summer snow (~ 25 ‰), this could be related to transport of external nitrate as suggested by Kunasek et al. (2008). Indeed, unlike at summer Summit conditions, nitrate transported from outside of the Arctic would be formed by both night and day time reactions and should possess higher $\Delta^{17}O$ than locally formed nitrate which is mainly from OH oxidation (Kunasek et al., 2008). In our model, the $\Delta^{17}O(NO_3^-)$ of $F_{pri}$ was assumed to be 30 ‰. Although this is unlikely to be the true value of long range transported nitrate, it can be viewed as the starting value and from which we can assess the effects of post-depositional processing (i.e., the changes caused by post-depositional processing) that is the focus of this study. In the model, the summer deposited nitrate possesses $\Delta^{17}O$ that is 1.9 ‰ lower than that of $F_{pri}$, due to the mixing of $F_{pri}$ with snow-sourced nitrate. In wintertime, local nitrate formation in the overlying atmosphere is muted in the model as there is no sunlight, and thus the deposited $\Delta^{17}O(NO_3^-)$ is completely controlled by $\Delta^{17}O(NO_3^-)$ of $F_{pri}$.

In addition, the cage effect during photolysis further reduces $\Delta^{17}O$ in snow nitrate by ~ 0.2 ‰. This is different from what occurs on the East Antarctic Plateau where the cage effect dominates the post-depositional $\Delta^{17}O(NO_3^-)$ decrease (Erbland et al., 2013). This is because on the East Antarctic Plateau, the snow accumulation rate is very low and nitrate remains in the photic zone for 5 years or longer (compared to less than a year at Summit, Greenland). Taking into account the cage effect in Summit snow, a 2.1 ‰ $\Delta^{17}O$ seasonality was simulated by the model, which is much smaller compared to the observed 9 ‰ seasonality (Figure 1c). Note as our model doesn't consider nitrate formation via $BrONO_2$ hydrolysis, which tends to produce nitrate with higher $\Delta^{17}O$ than OH oxidation, the modeled 2.1‰ seasonality is an upper limit. In all, the results suggest that post-depositional processing does not play a significant role in regulate the observed seasonality of $\Delta^{17}O(NO_3^-)$ at Summit,

which is probably mainly caused by seasonal differences in $\Delta^{17}O(NO_3^-)$ of $F_{pri}$ in addition to seasonal difference in local nitrate formations as suggested by Kunasek et al. (2008).

The observed surface snow $\delta^{15}N(NO_3^-)$ (green curve in Figure 1b) varies from –13.0 ‰ to –2.8 ‰ in a year (Jarvis et al., 2009). In comparison, observed snowpack $\delta^{15}N(NO_3^-)$ varies from (–9.8 ± 3.1) ‰ of the annual valleys to (6.3 ± 1.8) ‰ of the annual peaks (average of three years of observations) and displays apparent enrichments in spring and early summer. This difference suggests substantial changes in $\delta^{15}N(NO_3^-)$ after deposition. The model with constant $\delta^{15}N$ of $F_{pri}$ (i.e., 0 ‰ throughout the year) predicted a $\delta^{15}N(NO_3^-)$ seasonality with a spring peak (black dashed curve in Figure 2b), and the modeled

magnitude of seasonal difference is ~17.5 ‰ that is similar to the observations (16.1 ± 3.6) ‰ seasonality). But there is a constant model-observation discrepancy that the lowest $\delta^{15}N(NO_3^-)$ value in a year appears earlier in the model than in the observations. When including seasonal variations in $\delta^{15}N$ of $F_{pri}$ (i.e., using year-round surface $\delta^{15}N(NO_3^-)$), the modeled seasonal $\delta^{15}N(NO_3^-)$ pattern as well as the magnitude (~18.3 ‰) (blue curve in Figure 1b) became almost identical to the observations, except that the absolute values of the modeled $\delta^{15}N(NO_3^-)$ are on average 5.2 ‰ lower than the observations.

This modelled underestimate could be due to the use of observed $\delta^{15}N$ of surface snow nitrate ((−6.2 ± 1.1) ‰ on average) which may underestimate $\delta^{15}N$ of $F_{pri}$. The $\delta^{15}N$ of surface snow nitrate is affected by input of snow-sourced nitrate depleted in $\delta^{15}N$ in the summer. Therefore, the modeled snowpack $\delta^{15}N$ should be lower than the observation given that the starting values in the model are biased low. In comparison, the simulation with constant $\delta^{15}N$ of $F_{pri}$ (i.e., 0 ‰) predicted absolute values generally higher than the observations, which may be because the value of 0 ‰ might be an overestimate.

The occurrence of the spring $\delta^{15}N$ peak should be also driven by post-depositional processing. Post-depositional processing starts after polar sunrise and continues to operate until the beginning of polar winter. During this time, the effect of post-depositional processing accumulates, and the spring snow layer has experienced the largest degree of post-depositional loss and thus exhibits the most enriched $\delta^{15}N$. The annual snow thickness at Summit is ~ (65 ± 10) cm a$^{-1}$, which is twice the depth of the photic zone, and therefore there should be no additional post-depositional processing after a year and the spring

high $\delta^{15}N(NO_3^-)$ caused by post-depositional processing is preserved as seen in the model and observations.

**3.2 Seasonality of photolysis flux (FP) and deposition flux (FD)**

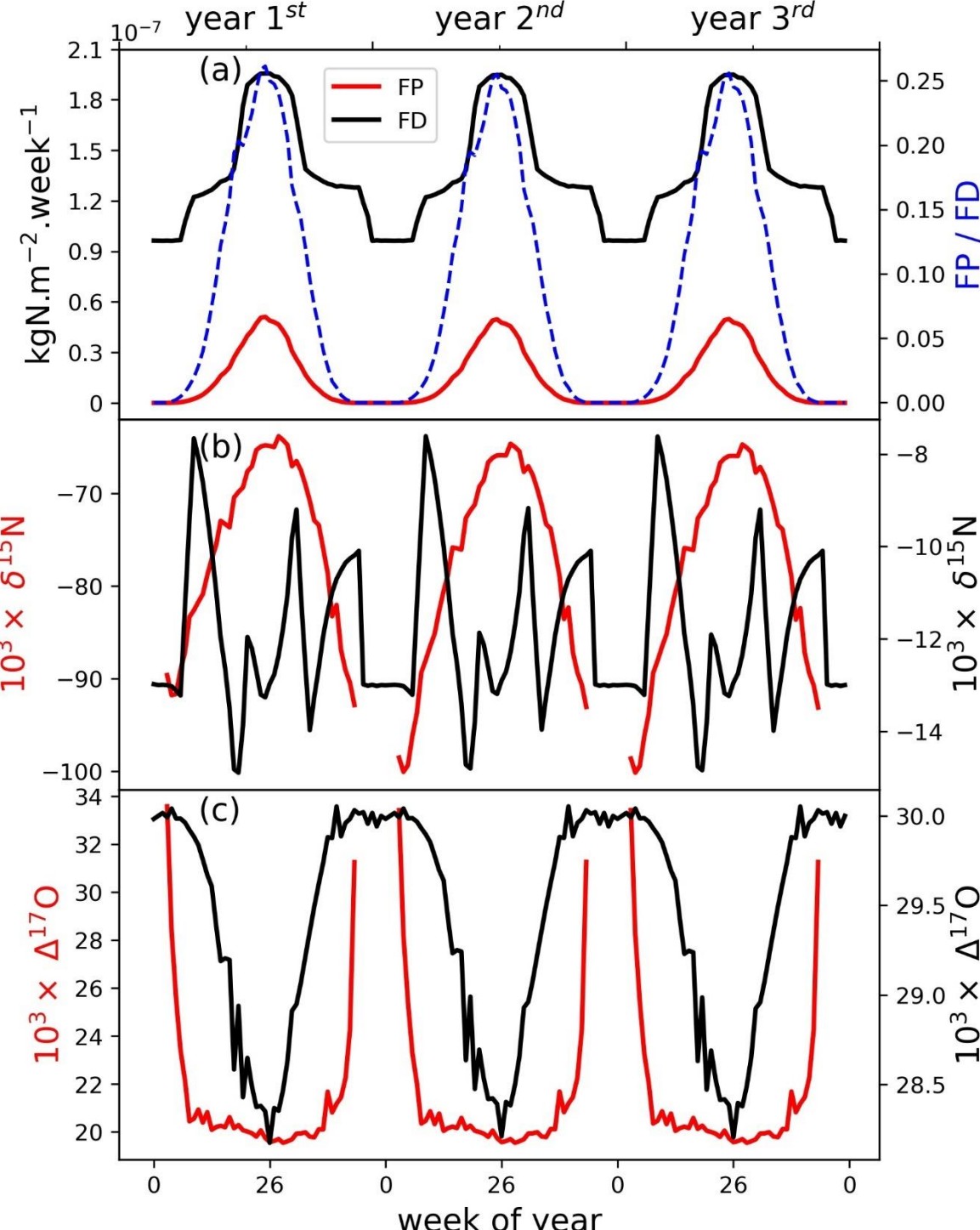

**Figure 2.** Weekly distribution of photolysis flux (FP) and deposition flux (FD) and their nitrate isotopic compositions. Gaps in FP (red line) are during the winter when there is no sunlight and thus no photolysis. The results shown are those simulated with seasonal variations in the flux and $\delta^{15}$N of primary nitrate ($F_{pri}$).

To discern the processes leading to the seasonal isotope patterns, we further investigated the weekly nitrate deposition flux (FD) and isotopes, as well as the weekly flux of snow-sourced nitrate (FP) and isotopes using the model. As shown in Figure 2a, during mid-summer when actinic flux reaches its maximum, FP reaches the maximum (and is zero in winter). Our simulated average daily $NO_2$ flux from snowpack in summer was $2.96 \times 10^{12}$ molecules m$^{-2}$ s$^{-1}$, in good agreement with summer observations at Summit ($2.52 \times 10^{12}$ molecules m$^{-2}$ s$^{-1}$, Honrath et al., 2002). FD is a mixture of $F_{pri}$ and FP, so it also reaches the maximum in summer due to the contribution of FP, in addition to the summer high $F_{pri}$. This at least in part explains the modeled summer nitrate concentration maximum. But even in summer, FP was only about 25 % of FD, demonstrating the importance of $F_{pri}$ in determining the budget of snow nitrate at Summit.

The $\delta^{15}$N of FP in summer half year (–77 ‰ to –65 ‰) was severely depleted compared to $F_{pri}$ (–6.7 ‰ to –2.8 ‰). As shown in Figure 2b, $\delta^{15}$N of FP gradually increased from the onset of photolysis, and reached the highest in mid-summer and decreased after that. This is mainly caused by the wavelength-dependent $\varepsilon_p$ (Berhanu et al., 2014) which varies from −57 ‰ to −87 ‰ and peaks in mid-summer at Summit (Figure 3a), corresponding to the smallest isotope effect in mid-summer. The $\delta^{15}$N($NO_3^-$) of FD was a combination of FP and $F_{pri}$. Therefore, a clear decrease in $\delta^{15}$N($NO_3^-$) of FD can be expected in summer (Figure 2b) when the contribution of FP was the largest. The isotope effect in $\delta^{15}$N during the deposition of nitrate was also included in the model but turns out to have no apparent impact on the modelled snowpack $\delta^{15}$N($NO_3^-$) profile. This is because that essentially all nitrate in the atmosphere except the fraction being exported was deposited (i.e., FD) over the period of each simulation step (i.e., one week), and thus the isotope effects were negligible due to mass balance.

The modelled $\Delta^{17}$O($NO_3^-$) of FP is mainly determined by local atmospheric chemistry, e.g., the NO-$NO_2$ cycling and the subsequent formation of $HNO_3$. Under the prescribed Summit atmospheric conditions, we calculated the $\Delta^{17}$O($NO_3^-$) of FP with a mean of (19.7 ± 0.3) ‰ during summer. This $\Delta^{17}$O($NO_3^-$) of FP combined with $F_{pri}$ ($\Delta^{17}$O = 30 ‰), leading to a summer minimum $\Delta^{17}$O of FD that was 1.9 ‰ lower than that of $F_{pri}$. An additional ~ 0.2 ‰ difference was induced upon archival from the cage effect, suggesting the cage effect plays a negligible role on snow $\Delta^{17}$O ($NO_3^-$) at Summit. This reinforces why oxygen isotopes should not be used to investigate the degree of post-depositional processing, especially at high snow accumulation sites, because it is dominated by regionally and/or local atmospheric processes.

In addition, our model results indicate apparent recycling of nitrate at the air-snow interface leading to changes in snow nitrate isotopes. This is opposite to Fibiger et al. (2016) who concluded there is little to no local recycling of nitrate at Summit based on the fact that surface snow nitrate $\Delta^{17}$O was not elevated when atmospheric BrO concentration increased. However, high BrO concentration does not necessarily lead to high atmospheric and/or snow nitrate $\Delta^{17}$O for several reasons. First, the production of BrO will consume $O_3$ and this is a tradeoff in terms of influencing $\Delta^{17}$O($NO_3^-$). Second, BrO concentration always covary with HOx (HOx = OH, $RO_2$ and $HO_2$) (Liao et al. 2011), and increases in HOx tend to lower $\Delta^{17}$O($NO_3^-$) that

offsets the effect of increased BrO concentration on $\Delta^{17}O(NO_3^-)$. Third, the observed BrO increase by Fibiger et al. (2016) only lasted for a few hours, and whether this is long enough to significantly perturb the local atmospheric nitrate budget over longer time periods and nitrate in snow is questionable. Nevertheless, BrONO₂ hydrolysis is only one of the many pathways of atmospheric nitrate formation, and for atmospheric nitrate to perturb the surface snow nitrate budget (in the top 2 cm, the observations by Fibiger et al. (2016)) through dry deposition it will need days to weeks of nitrate deposition given its dry deposition flux ($7.16 \times 10^{11}$ molecules m$^{-2}$ s$^{-1}$) at Summit (Honrath et al., 2002). A chemical transport model with the post-depositional processing incorporated would be best to investigate this further but is out of the scope of this study.

### 3.3 Loss of snow nitrate due to photolysis at Summit

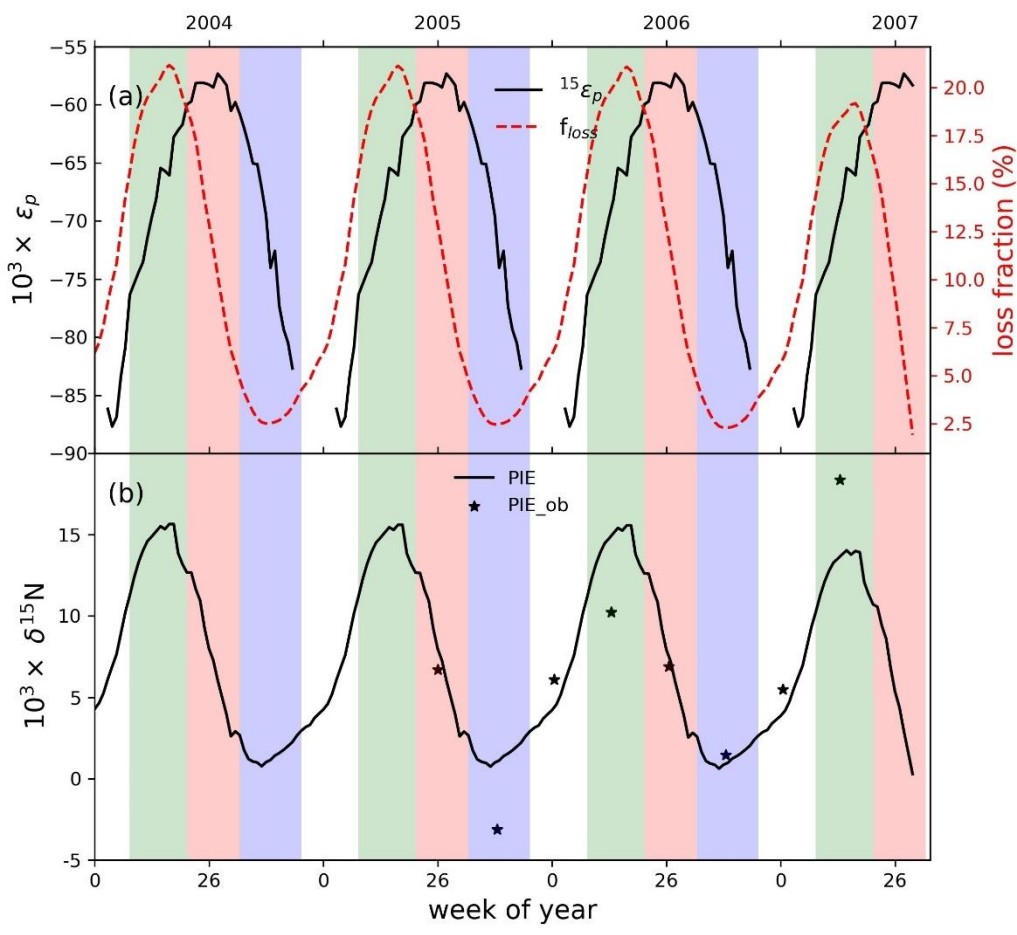

**Figure 3.** (a) The fraction of nitrate loss after deposition and the photolysis fractionation factor ($\varepsilon_p$) at different weeks. (b) PIE: the photo-induced isotope effect. The solid star represents the estimated PIE from surface and snowpack nitrate data reported

by Jarvis et al. (2009). The green, red, blue and white background color represents spring, summer, autumn and winter, respectively.

The lost fraction ($f_{loss}$) of snow nitrate upon archival is plotted in Figure 3a, calculated as the difference in nitrate concentration of an archived layer to the concentration when it was at the surface. As shown in Figure 3a, throughout a year, $f_{loss}$ varied from 1.9 % to 21.1 %, similar to the < 7 % to 25 % loss estimated by Burkhart et al. (2004) and Dibb et al. (2007). In particular, Dibb et al. (2007) calculated the average $NO_3^-$ concentrations in fresh and buried snow layers, and found a mean of ~ 9 % loss which is in good agreement with our calculated mean $f_{loss}$ of (10.4 ± 6.6) %. The loss of nitrate in a snow layer corresponds to the enrichment of $\delta^{15}N(NO_3^-)$ in that layer. Here we defined the enrichment in snow $\delta^{15}N(NO_3^-)$ due to photolysis as PIE (the photo-induced isotope effect), i.e., the difference between $\delta^{15}N(NO_3^-)$ of a newly deposited snow layer and the same layer that was finally buried below the photic zone. As shown in Figure 3b, PIE is the highest in the 18[th] week of the year, corresponding to the time of the highest $f_{loss}$. In addition, PIE displays a maximum in spring and minimum in autumn, in good agreement with the observed seasonal $\delta^{15}N(NO_3^-)$ pattern in snowpack. We also estimated PIE based on the observed $\delta^{15}N(NO_3^-)$ in surface snow and snowpack at Summit as reported by Jarvis et al. (2009). As shown in Figure 3b, PIE estimated based on observations (PIE_ob) agrees well with the modelled PIE. These further confirm the dominant role of the photo-driven post-depositional processing in the seasonal snowpack $\delta^{15}N(NO_3^-)$ pattern. Note in the model neither $f_{loss}$ nor PIE varied with seasonal differences in the flux and $\delta^{15}N$ of $F_{pri}$, respectively. The agreement between the modeled and observed PIE further demonstrates the dominant role of photo-driven post-depositional processing in seasonal $\delta^{15}N(NO_3^-)$ variations at Summit. Physical loss of snow nitrate through adsorption/evaporation is associated with very small $\varepsilon_p$ (e.g., 3.6 ‰ by Erbland et al. (2013)), and given the mean summertime temperature at Summit (261 ± 3 K) and a maximum lost fraction of 25 % (Dibb et al., 2007), only a ~1 ‰ change in $\delta^{15}N(NO_3^-)$ can be caused by physical loss.

The $f_{loss}$ calculated above was referred to a specific archived layer relative to when it was at the surface, and part of the loss was recycled to layers above that specific layer. Therefore, the net loss integrated over a certain period should be less than $f_{loss}$. Here we calculated an annual net loss $f_{loss}$ as follows:

$$\bar{f}_{loss_{annual}} = 1 - \frac{F_a}{F_{pri}}$$

(6)

where $F_a$ represents the archival flux of nitrate ($6.33\times10^{-6}$ kgN m$^2$ a$^{-1}$), and $\bar{f}_{loss\_annual}$ was calculated as 4.1 %. This is consistent with the annual mean $\delta^{15}N(NO_3^-)$ which was 2.6 ‰ enriched compared to $\delta^{15}N$ of $F_{pri}$. For $\Delta^{17}O(NO_3^-)$, upon archival, the annual mean is 0.9 ‰ lower than $\Delta^{17}O$ of $F_{pri}$. These values represent the integrated effects of the post-depositional processing on isotopes of the archived nitrate under present Summit conditions. In addition, these results suggest that although photochemistry was active and resulted in significant redistribution of snow nitrate in the photic zone at Summit, the annual net loss is small, consistent with the results of previous studies at Summit based on cumulative inventory assessment of nitrate mass in snowpits (Burkhart et al., 2004; Dibb et al., 2007), as well as the result from a south-eastern Greenland ice core where negligible annual nitrate loss was suggested due to the even higher snow accumulation rate (≈ 300 cm snow per year) than

Summit (≈ 65 cm snow per year) (Iizuka el al., 2018). It is also interesting to note that despite having a similar source region (Geng et al., 2015, Iizuka el al., 2018), $\delta^{15}N(NO_3^-)$ in this south-eastern Greenland ice core is lower than in Summit ice cores (personal communication with Shohei Hattori). This is qualitatively consistent with the difference in the snow accumulation rate at the two sites, since the lower snow accumulation rate at Summit will result in higher degree of post-depositional processing.

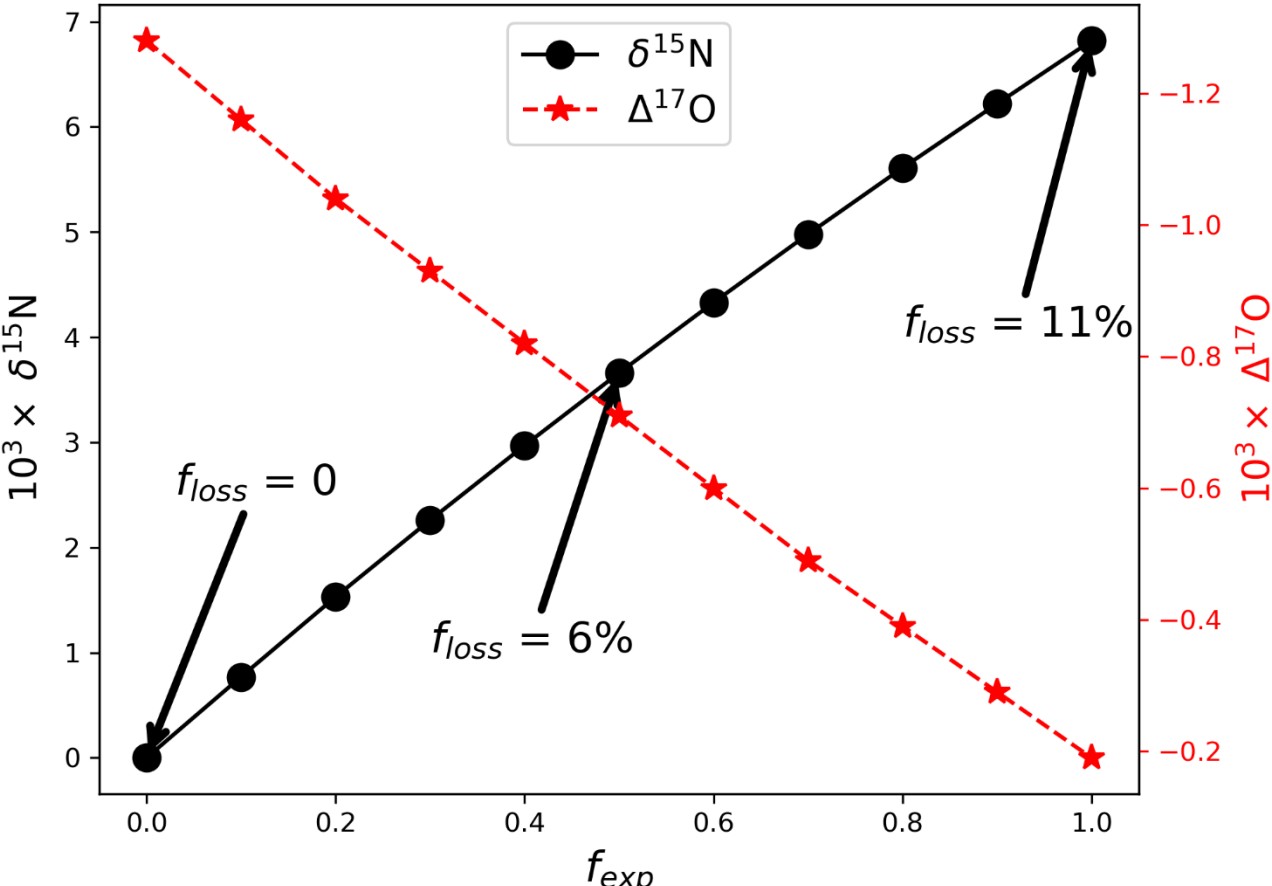

**Figure 4.** Sensitivity of annual mean $\delta^{15}N(NO_3^-)/\Delta^{17}O(NO_3)$ upon archival to $f_{exp}$. Positive/negative values indicate the deviations to $F_{pri}$. Note when $f_{exp}$ is set to 1, the small non-zero value (−0.19 ‰) of $\Delta^{17}O(NO_3^-)$ represents the effects of the cage effect.

The annual net loss in the model is mainly determined by $f_{exp}$ which represents the fraction of exported nitrate from the site of photolysis. Although $f_{exp}$ doesn't influence the loss fraction of a specific snow layer and subsequently the predicted seasonal $\delta^{15}N(NO_3^-)$ pattern as modeled (Supplementary Figure S3), it determines how much of the reformed nitrate was

recycled back to snow. In Figure 4, we investigated the sensitivity of the annual net loss, and the annual mean archived $\Delta^{17}O(NO_3^-)$ and $\delta^{15}N(NO_3^-)$ to $f_{exp}$. We found the archived $\Delta^{17}O(NO_3^-)$ decreases with increasing $f_{exp}$ while $\delta^{15}N(NO_3^-)$ is the opposite, because larger $f_{exp}$ corresponding to less contribution of FP to FD. Under the extreme circumstance with $f_{exp} = 1$, i.e., all snow-sourced nitrate was exported, $\delta^{15}N(NO_3^-)$ in snow was on average 6.8 ‰ enriched compared to primary $F_{pri}$ under

present Summit conditions, while $\Delta^{17}O(NO_3^-)$ was only 0.2 ‰ lower than $\Delta^{17}O$ of $F_{pri}$ caused entirely by the cage effect. In this study, $f_{exp}$ was determined to be 0.35 following the method used by Erbland et al. (2015), but this could be an underestimate. At Summit, observations by Honrath et al. (2002) indicate that $NO_x$ and/or $HNO_3$ emitted from sunlit snow are largely exported from the local boundary layer if no wet deposition occurs.

### 3.4 Implications for interpretation of ice core nitrate isotope records

Due to the fast cycling of nitrate at the air-snow interface, the annual net loss (4.1 %) and the associated annual mean changes in $\delta^{15}N(NO_3^-)$ (2.6 ‰) and $\Delta^{17}O(NO_3^-)$ (0.9 ‰) caused by post-depositional processing are small under present Summit conditions. Despite this, at seasonal scale, given the strong variations in actinic flux, post-depositional processing plays an important role in the seasonal $\delta^{15}N$ fluctuation. The degree of post-depositional processing is also strongly depending on snow accumulation rate which is usually very different in different climates. As such, the net loss and the associated isotope effects

could be increased in periods with a reduced snow accumulation rate. For example, over the last glacial-interglacial period, considering only the changes in snow accumulation rate at Summit (Geng et al., 2015), the model calculated a 11 % annual nitrate loss in the glacial period and a glacial-interglacial $\delta^{15}N$ difference of 9.2 ‰. In comparison, the observed glacial-interglacial $\delta^{15}N$ difference is (16.7 ± 4.8) ‰ (Geng et al., 2015). This suggests changes in the degree of post-depositional processing caused by the glacial-interglacial snow accumulation rate difference alone can explain more than half of the

observed $\delta^{15}N(NO_3^-)$ difference. Note the modeled 11% net loss in the glacial climate according to Equation (2) is not in conflict with the (45-54) % loss estimated by Geng et al. (2015) who calculated the loss fraction from $F_a$ and FD instead of $F_{pri}$. If replacing $F_{pri}$ in Equation (2) with FD, the loss fraction is then 31 %. With the effects of changes in snow accumulation rate alone, the model predicted the glacial $\Delta^{17}O(NO_3^-)$ would be 2 ‰ lower than in the present. This amount is significant compared to the observed glacial-interglacial $\Delta^{17}O(NO_3^-)$ difference of 6.2 ‰ (Geng et al., 2017). Note there are many other

factors can influence the degree of post-depositional processing in the glacial climate, e.g., local wind speed, actinic flux, quantum yield of snow nitrate photolysis, and etc., which are out of the scope of this study. But our results here reinforce the effects of post-depositional processing on ice-core nitrate concentrations and isotopes even at high snow accumulation rate sites, and such effects must be quantified and corrected in order to use ice-core nitrate records to retrieve past information on $NO_x$ emissions and abundance and atmospheric oxidation capacity especially when the records cover different climates.

# 4 Conclusions

In this study we applied the TRANSITS model to explicitly investigate the impact of the photo-driven post-depositional processing on the preservation of nitrate and its isotopes at Summit Greenland, with the focuses on changes in nitrate isotopes after deposition. The results suggest that the photo-driven post-depositional processing is active at Summit, causing strong redistribution of snow nitrate accompanied by isotope effects in the photic zone. Despite the high snow accumulation rate at Summit, up to 21 % loss/redistribution of nitrate can be induced by the photolysis, resulting in a spring $\delta^{15}N(NO_3^-)$ peak consistent with the observations. Despite uncertainties in the model, e.g., specific surface area, quantum yield of snow nitrate photolysis, the export fraction, the modeled loss/redistribution of nitrate after deposition is consistent with previous studies, and explains the observed difference between $\delta^{15}N(NO_3^-)$ in surface snow and snow at depth. The latter is evidence of changes in $\delta^{15}N(NO_3^-)$ after deposition. The model also reproduced the observed seasonal patterns of snow nitrate concentration and $\delta^{15}N(NO_3^-)$ reasonably well, and the model-observation discrepancy in the timing of the lowest seasonal $\delta^{15}N(NO_3^-)$ was addressed when seasonal variations in $\delta^{15}N(NO_3^-)$ of $F_{pri}$ was included. But the effects of $\delta^{15}N$ of $F_{pri}$ on snow $\delta^{15}N(NO_3^-)$ seasonality appear to be mainly pronounced in autumn/winter, i.e., the period with the lowest seasonal $\delta^{15}N(NO_3^-)$ when photolysis is negligible. This makes sense as when photolysis is muted snow nitrate $\delta^{15}N(NO_3^-)$ should be the same as that of $F_{pri}$. When photolysis is active, the $\delta^{15}N(NO_3^-)$ signal of $F_{pri}$ is not preserved. In contrast, the post-depositional processing only led to 2.1 ‰ seasonal change in $\Delta^{17}O$. These results are consistent with the expectation that photo-driven post-depositional processing modifies $\delta^{15}N$, but has only moderate impacts on $\Delta^{17}O$.

Overall, the model results suggest an important, perhaps even dominant role of post-depositional processing in regulating the snowpack $\delta^{15}N(NO_3^-)$ seasonality at Summit. Although the impact of photolysis of snow nitrate on $\delta^{15}N(NO_3^-)$ must be carefully evaluated when interpreting snowpack and ice core nitrate isotopes records even at sites with high snow accumulate rate such as Summit, Greenland, we note that this study does not address to what extent seasonal variations in $\delta^{15}N(NO_3^-)$ of $F_{pri}$ affect the snowpack $\delta^{15}N(NO_3^-)$. Observations on the concentration and isotopic composition of $F_{pri}$ and its seasonal variations would be best to answer this question. However, it would be difficult to distinguish primary from recycled nitrate during sunlit time periods due to the local influence of snow-sourced nitrate. Additional observations including a full year or multiple years of atmospheric nitrate isotopes along with surface snow and snowpack data at a single site should be pursued in the future to fully investigate the evolution of nitrate isotopes before and after deposition, and to thoughtfully evaluate the effects of post-depositional processing. On the other hand, precise measurements of snowpack properties, e.g., specific surface area, impurity concentrations, observational constraints on the quantum yield of snow nitrate photolysis, and better constraints on the export fraction are also needed in order to improve the model's performance.

*Code availability.* The codes for the numerical simulations and their analysis will be provided upon direct request to the corresponding author.

*Author contributions.*

480    L.G conceived this study; Z.J. performed the model simulations, analyzed the data, and wrote the manuscript with L.G. J.E., J. S. and B.A. provided the model and helped with the model setup. All authors contributed to data interpretation and writing.

*Competing interests.* The authors declare that they have no conflict of interest.

*Acknowledgements:* L.G. acknowledges financial support from the National Natural Science Foundation of China (Awards: 41822605 and 41871051), the Fundamental Research Funds for Central Universities, the Strategic Priority Research Program of Chinese Academy of Sciences (XDB 41000000), and the National Key R&D Program of China (2019YFC1509100). This work was partially supported by the French national programme LEFE/INSU, the ANR grants ANR-15-IDEX-02

(project IDEX Université Grenoble Alpes) and ANR-16-CE01-0011-01 (EAIIST project) of the French Agence Nationale de la Recherche (J.S. and J.E.). The French Polar institute (Institut Polaire Français - program SUNITEDC 1011) is thanked for field and funding support (JS, JE). B. A. acknowledges support from NSF (award PLR 1542723). Z. J. thanks John Robinson for his assistance in starting the TRANSITS model. The model outputs data are available through OSF, the Open Science Framework (http://doi.org/ 10.17605/OSF.IO/A4B7D).

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
