# Peer review of "Impacts of the photo-driven post-depositional processing on snow nitrate and its isotopes at Summit, Greenland: a model-based study"

_The Cryosphere, 2021_

## Author Comment (AC1)

We appreciate the reviewer for the time and efforts to review this manuscript, and for the suggestions/comments which improve the manuscript significantly. Below we list detailed responses to the reviewer's suggestions and comments. The comments are listed in italics, followed by the response in normal font.

**Comments**

*In this study, the authors carried out a model study, trying to reveal the post-depositional processes of snow nitrate and isotopes at Summit, Greenland. This study addressed the question for the snow nitrate regarding the ice core study and in the scope of The Cryosphere.*

*The model was proposed by Erbland et al. (2015) and has been applied for the investigation of the post-depositional process of snow nitrate in Antarctica. The field data used in the model was taken from the previous studies. The present study demonstrated the significant redistribution of nitrate in the upper snowpack due to the photolysis in the high accumulation site. In addition, the effects of the post-depositional process on the isotopes ($\delta^{15}N$ and $\Delta^{17}O$) were investigated in a quantitative way. Thus, the present study has novelty and impact to be published in this journal after revising.*

*The methods were clearly written and suitable for this study. However, some assumptions in the model were not discussed, such as the effect of the wavelength, the wind blowing, the temperature, and the evaporation as mentioned in the specific comments. In addition, there is a lack of evaluation of the present study comparing to the previous model studies as mentioned in the specific comments.*

**Response**: Thank you for the comments, we responded in details in the specific comments below.

Specific comments

*Line 60-63: This sentence was supported by field observations (Erbland et al. 2013; Noro et al. 2018)*

**Response**: Thanks for this suggestion. We have added these references in our main text accordingly.

*Line 95-96: The e-folding depth depends on the wavelength (Noro and Takenaka 2020). How did you obtain the e-folding depth of each wavelength from 280-350 nm?*

**Response**: Thank you this point. The e-folding depth does vary with wavelength, but here in the text we only report the value at 305nm to compare with other results at the same wavelength. To get the e-folding depth from 280nm to 350 nm, we used the method as follows: the attenuation of actinic flux in snowpack was modelled by using a two-stream snow radiative transfer model TARTES (Libois et al., 2014). TARTES calculated the radiative transfer by using the specified snow properties such as LAI contents, SSA, snow grainsize, snow density etc. The wavelength-dependent inherent optical properties (extinction coefficients $\sigma_{ext}$ and absorption coefficients $\sigma_{abs}$) was calculated according to the complex refractive index of impurities and ice. The e-folding depth **for each wavelength** was calculated according to the normalized

actinic profile and the result in 305 nm was chosen to compare with previously published results at Summit. The calculated e-folding depth showed a gradually increasing trend towards longer wavelength, similar to the filed observation at Summit (Galbavy et al., 2006) but in contrast with Noro and Takenaka. (2020). We have made this process more clearly by adding additional details in the revised manuscript.

*Line191-192: Mean values of the accumulation data were used to avoid the negative values induced by the wind blowing in the present study. However, Pham et al. (2019) demonstrated that the wind blowing dominates the removal of the photodegradable organic contaminants from the surface snow in Antarctica (Pham et al. 2019). Therefore, the effect of the wind blowing should be discussed (I do not mean that authors have to conduct the model study which includes the wind blowing.).*

**Response**: We thank the reviewer for this comment. But we think these are two different scenarios. "Wind blowing" in our manuscript was referred to wind blowing snow that remove both snow and nitrate in it, when such thing occurs, both snow nitrate and its products leaves, and it should have null effects on the leftover. Since here are only two weeks among 52 weeks that displayed slightly negative accumulation rate, we think the impact of using average value should be minor.

*Line 238: Jarvis et al. (2009) reported the surface snow $\delta^{15}N$ $(NO_3^-)$ only for 5 months (March to July). How did you obtain the annual data? In addition, if you have each data point of Jarvis et al. (2009), please indicate in the same manner as observation (plots and lines) in Fig. 1.*

**Response**: Jarvis et al. (2009) also reported seasonal mean $\delta^{15}N(NO_3^-)$ of surface snow samples covering two years as listed in Table 2 of the that paper. These were what we used to get the annual average as well as seasonal variations that used to constrain the model for sensitivity test.

*Line 274-275: please add citations for the wavelength dependent of $\varepsilon_p$.*

**Response**: Thanks for pointing this, we have added Berhanu et al. (2014).

*Line 274-275: Does this sentence means that the wavelength change in season affects the $\varepsilon_p$, resulting in the peak of the FP $(\delta^{15}N)$ in mid-summer? In this case, please show the data for the wavelength change.*

**Response**: Yes, as modeled (Frey et al., 2009) and experimental determined by Berhanu et al. (2014), $\varepsilon_p$ is sensitive to wavelength. This is why in Figure 3a the calculated $\varepsilon_p$ varies seasonally from -60 to -90 per mil as the spectra reaching surface differs at different time.

*Fig. 2: Please explain why the FD $(\delta^{15}N)$ is changing.*

**Response**: FD is a mixture of $F_{pri}$ and the snow-sourced nitrate (FP). In the model, $\delta^{15}N$ of $F_{pri}$ varied seasonally, and the $\delta^{15}N$ of FP also varied seasonally due to the variations in $\varepsilon_p$. In addition, the relative contributions of Fpri and FP to FD were also

different at different time. So $\delta^{15}N$ of FD varied with time. This has been discussed in the manuscript (the second paragraph of section 3.2).

*In regard to evaporation/volatilization: The effect of evaporation was neglected in the present study. Shi et al. (2019) demonstrated that 38% of nitrate was lost from the snow sample at –4â for 14–16 days (Shi et al. 2019). Moreover, the temperature of the surface snow is closed to 0â in the daytime in summer in the Antarctic coastal site (Noro et al. 2020). Thus, the potential impacts of evaporation should be discussed in the present study.*

**Response**: We thank the reviewer for this comment. But we don't think evaporation is significant. The Shi et al. 2019 experiment was not a fair design. They collected Dome A snow and put it in an open-door room at Zhongshan Station (a coastal site). Dome A snow is of much higher concentrations than in the coast (500 ppb at Dome A while 30 ppb at the coast, Shi et al., 2015), a re-equilibrium between snow and the overlying air will be established which artificially enhances the physical release. In addition, Erbland et al. 2013 suggested that the apparent $\varepsilon_p$ became closer to zero at higher snow accumulation rate site is due to enhanced role of evaporation. However, this conclusion is questionable as the derived apparent $\varepsilon_p$ is also influenced by snow nitrate that has not experienced photolysis, the higher of this fraction (at high accumulation rate site), the closer $\varepsilon_p$ to zero.

What is more, in both evaporation experiments (including Erbland et al. (2013)), when mass loss is significant at higher temperature, the fractionation factor is very small. At Summit, even using a 25 % mass loss (the maximum loss fraction, given by Dibb et al. (2007)), and a $\varepsilon_p$ of 3.6 ‰ (Erbland et al., 2013), only a 1 per mil difference in $\delta^{15}N$ can be induced.

*In regard to the positioning of the model compared to the previous studies:*
*The model studies have been reported, related to the post-depositional process of nitrate in Greenland (e.g. Zatko et al. 2016). The advantages and the disadvantages of the models proposed in the previous studies and the present study should be described to demonstrate the positioning of the present study as a paragraph in the Introduction or as a section in the Results and discussion.*

**Response**: Thank you. The Zatko et al. (2016) study using a global 3-D chemical transport model (GEOS-Chem model). But the model treated snowpack as a whole and didn't specify the behaviors of nitrate at different depths in the photic zone, and can't distinguish seasonal differences. In addition, it didn't incorporate isotope fractionation associated with photolysis, but instead using a fixed fractionation constant and a Rayleigh fractionation model to calculate the changes in isotope with mass loss. While the TRANSITs model is a layer specific model and treats the reaction and recycle of nitrate step-by-step, and it can predict the changes of isotopes in each defined layer (e.g., the seasonal changes). We have added this briefly in the introduction.

*Technical corrections*
*Line 20 and any other pars: Space is not needed before "%" and "‰".*
**Response**: Thanks for this comment, but according to IUPAC recommendation, the dimension quality of physical quantities such as "%" and "‰" should be treated as units, thus a space is necessary when writing "%" and "‰" after a number.

*Line 32 and many other parts: "Minus" should not be indicated as "-" but "−".*
**Response**: Thank you and we have revised in the manuscript accordingly.

*Line 110: $J(NO_2) \rightarrow J_{(NO2)}$*
**Response**: Revised accordingly.

*Line 215: won't→will not*
**Response**: Revised accordingly.

*Line 279: ware null→were negligible*
**Response**: Revised accordingly.

*Fig. 3: Please spell out $F_{pri}$ in the caption.*
**Response**: Did you mean in Figure 2? We have spelled it out.

*Fig. 3: Remove the frame border of the legend.*
**Response**: Revised accordingly.

**Reference**

Berhanu, T. A., Meusinger, C., Erbland, J., Jost, R., Bhattacharya, S., Johnson, M. S., & Savarino, J.: Laboratory study of nitrate photolysis in Antarctic snow. II. Isotopic effects and wavelength dependence, J. Chem. Phys., 140, 244306, https://doi.org/10.1063/1.4882899, 2014.

Dibb, J. E., Whitlow, S. I., & Arsenault, M.: Seasonal variations in the soluble ion content of snow at Summit. Greenland: Constraints from three years of daily surface snow samples, Atmos. Environ., 41, 5007-5019, https://doi.org/10.1016/j.atmosenv.2006.12.010, 2007.

Erbland, J., Vicars, W., Savarino, J., Morin, S., Frey, M., et al.: Air–snow transfer of nitrate on the East Antarctic Plateau-Part 1: Isotopic evidence for a photolytically driven dynamic equilibrium in summer, Atmos. Chem. Phys., 13, 6403-6419, https://doi.org/10.5194/acp-13-6403-2013, 2013.

Frey, M. M., Savarino, J., Morin, S., Erbland, J., & Martins, J.: Photolysis imprint in the nitrate stable isotope signal in snow and atmosphere of East Antarctica and implications for reactive nitrogen cycling, Atmos. Chem. Phys., 9, 8681-8696, https://doi.org/10.5194/acp-9-8681-2009, 2009.

Galbavy, E. S., Anastasio, C., Lefer, B. L., & Hall, S. R.: Light penetration in the snowpack at Summit, Greenland: Part 2: Nitrate photolysis, Atmos. Environ, 41, 5091-5100, https://doi.org/10.1016/j.atmosenv.2006.01.066

Jarvis, J. C., Hastings, M. G., Steig, E. J., & Kunasek, S. A.: Isotopic ratios in gas-phase $HNO_3$ and snow nitrate at Summit, Greenland, J. Geophys. Res. Atmos, 114, D17301,

https://doi.org/10.1029/2009JD012134, 2009.

Libois, Q., Picard, G., France, J., Arnaud, L., Dumont, M., Carmagnola, C., & King, M.: Influence of grain shape on light penetration in snow, The Cryosphere, 7, 1803-1818, https://doi.org/10.5194/tc-7-1803-2013, 2013.

Shi, G., Chai, J., Zhu, Z., Hu, Z., Chen, Z., et al.: Isotope fractionation of nitrate during volatilization in snow: a field investigation in Antarctica, Geophys. Res. Lett., 46, 3287-3297, https://doi.org/10.1029/2019GL081968, 2019.

Zatko, M., Geng, L., Alexander, B., Sofen, E., & Klein, K.: The impact of snow nitrate photolysis on boundary layer chemistry and the recycling and redistribution of reactive nitrogen across Antarctica and Greenland in a global chemical transport model, Atmos. Chem. Phys., 16, 2819-2842, https://doi.org/10.5194/acp-16-2819-2016, 2016.

---

## Author Comment (AC2)

We appreciate the reviewer for the time and efforts to review this manuscript, and for the suggestions/comments which improve the manuscript significantly. Below we list detailed responses to the reviewer's suggestions and comments. The comments are listed in italics, followed by the response in normal font.

**Comments**

*Jiang et al. revisited previously published Summit snowpack nitrate isotope data using a snow photochemical column model (TRANSITS). It was found that post-depositional processes at Summit can explain the observed seasonal variability of d15N without considering the source variability of reactive nitrogen species. Although the modeling result may be further tested and improved by observation-based parameterizations of some parameters such as d15N of atmospheric nitrate, this work challenges previous expectations and highlights that post-depositional processes on isotopic compositions of cryospheric nitrate at a site with relatively high snow accumulation rates like Summit still should be carefully considered in the future. The model is scientifically sound and its uncertainties are well discussed. I expect that the findings will stimulate more studies. I recommend its publication in The Cryosphere and only have some minor suggestions to improve the clarity of this manuscript.*
**Response**: Thank you for the comments.

*Lines 30-35: I would suggest the authors to introduce d15N as well because d15N is the major focus of this study.*
**Response**: Thanks for this suggestion. We have added such text and relevant references in the introduction part to illustrate the importance of snow/ice core $\delta^{15}N(NO_3^-)$.

*Line 49: As stated later by the authors, "the degree of post-depositional processing and the induced effects on snow nitrate and isotopes vary site by site" (Line 56). The values presented here seem too precise to represent the fractionation factors "under typical polar conditions".*
**Response**: We removed the use of "under typical polar conditions" and specified it as Dome C local conditions. This is just to show how large the fractionation factor could be.

*Lines 51-55: The local atmospheric chemistry of photolysis product NO₂ as stated earlier (lines 42-47) may also alter D17O values.*
**Response**: Thanks for pointing this. We added a sentence to illustrate this point as follows: "On the other hand, the photoproducts of snow nitrate released into local atmosphere would rapidly reform nitrate and redeposit in situ, thus imprinting the snow $\Delta^{17}O(NO_3^-)$ with local atmospheric oxidizing conditions."

*Lines 60-63: The logic may be more clear if the authors can give the snow accumulation rate at Summit and discuss whether post-depositional processes were expected based on this number.*

**Response**: We have added "Summit, Greenland is a typical high snow accumulation site (250 kg m$^{-2}$ a$^{-1}$, Dibb et al., 2004), and under this high snow accumulation condition nitrate would be preserved better compared to low snow accumulation sites such as at Dome C, but the magnitude remains unquantified."

*Lines 65: It may be better to quantify "surface" (<3 cm?) so that readers can compare the number to 30-40 cm without reading Fibiger's papers.*

**Response**: Fibiger et al. (2013) stated that their surface snow sample comprised the top 1-2 cm snow, while in Fibiger et al. (2016) they became 0.5-3 cm in depth. In the revised manuscript, we have added the overall range of their samples (<3 cm) as reference.

*Lines 82-89: The term "snow-sourced" nitrate appears many times in the rest of manuscript but not here. I would suggest the authors to define "primary" and "snow-sourced" nitrate at the beginning.*

**Response**: Thanks for this suggestion. We have introduced these two terms in the introduction.

*Line: 170: The value of k is not given.*

**Response**: It is 3 × 10$^{-12}$ cm$^3$ molecule$^{-1}$ s$^{-1}$ and has been added.

*Lines 178-182: It is unclear how the epsilon-d is determined and which value is used. (I suggest the authors to give a table, in either main text or supplementary materials to show all parameters and values used in the model and sensitivity tests).*

**Response**: The $\varepsilon_d$ (+10‰) was chosen from Erbland et al. (2013). The +13‰ difference from Fibiger et al. (2013) was not used here because this value was measured in late spring and could possibly perturbed by photolysis of snow nitrate. In our revised manuscript we have made the value clearer. We have also added a table in supplement file to show the parameters and values in our simulation for better reference. Thanks for this suggestion.

*Lines 264-271: There are too many terms that are similar but not clearly organized (FD, FP, Fpri, snow-sourced). They could have been more clear. I would suggest the authors to define everything at the very beginning (e.g., Section 2). Otherwise it may be difficult for readers to follow.*

**Response**: Thanks for this suggestion. In the revised manuscript, we defined $F_{pri}$ and snow sourced nitrate in the introduction, and the other two in sections where they first mentioned.

*Figure 2: Since FP is zero in winter, the authors may want to remove red lines during wintertime in panels b and c.*

**Response**: Thanks for this suggestion. We have revised the figures and removed the wintertime part of the $\delta^{15}N/\Delta^{17}O(FP)$ in our new figures.

*Lines 299: The definition is not clear. What is the baseline?*
**Response**: PIE refers to the difference between $\delta^{15}N(NO_3^-)$ of surface snow and the same layer that buried below photic zone. We have added this statement to make it more specified in our revised manuscript.

*There are some grammatical errors that need to be corrected (e.g. Lines 172, 230, 248).*
**Response**: Revised as suggested.

---

## Author Comment (AC3)

**Comments:**

*I think it is critically important that the authors do more to compare with the larger body of isotopic measurements of nitrate in the snow and atmosphere at Summit, Greenland. The conclusions drawn here focus on the loss and recycling of nitrate when the model can just as well explain the observations based on a change in the primary nitrate isotopic signal rather than photolytic redistribution of nitrate in the snow. A number of works from my research group on surface snow and snowpack at Summit, Greenland have all concluded that photolytic loss OR recycling cannot explain the observations presented in those studies. If the model here is indeed correct, this is an opportunity then to re-interpret those conclusions. Or better constrain the model and present stronger evidence-based conclusions. Simply put – the model has multiple tuning options to improve agreement between predictions and observations, but can it actually explain what's observed at Summit?*

**Response**: We appreciate Dr. Hastings's efforts reading and raising concerns about this manuscript. Before we respond to Dr. Hastings's specific comments below, we would like to clarify a few things:

1) Dr. Hastings may have misunderstood the purpose of this manuscript. This manuscript is explicitly focused on to what degree the magnitude of the observed seasonality can be explained by post-depositional processing, and the TRANSITS model we used here was not constructed to predict the isotopes of snow nitrate ($\delta^{15}N$ and $\Delta^{17}O$), but to assess their **changes** caused by post-depositional processing of nitrate after primary deposition. As shown by field observations and laboratory experiments, snow nitrate will be photolyzed as long as sunlight is available (Chu and Anastasio, 2003, Dibb et al., 1998; Honrath et al., 2002; Meusinger et al., 2014). **Therefore, given the significant differences in actinic flux between the summer half year and winter half year in the polar regions, seasonality in photochemistry and the degree of post-depositional processing should be questioned**. Our model reproduced the observed snowpack NOx flux in summer conditions at Summit (modeled $2.96 \times 10^{12}$ molecules m$^{-2}$ s$^{-1}$, vs. $2.52 \times 10^{12}$ molecules m$^{-2}$ s$^{-1}$ observed by Honrath et al., 2002), and it showed that the photo-driven post-depositional processing can explain the magnitude of the seasonality (difference between summer and winter) in $\delta^{15}N$, **no matter what $\delta^{15}N$ values were assumed for primary nitrate**. At the same time, the model cannot explain the magnitude of the seasonality in $\Delta^{17}O$. This is consistent with previous work in Antarctica, and just as expected: photolysis induces large N isotope fractionations, but won't directly influence $\Delta^{17}O$. Note changes in $\Delta^{17}O$ upon nitrate preservation observed in East Antarctic Plateau (where snow accumulation rate is very low) is mainly caused by cage effect, in addition to recycling of nitrate at the air-snow interface. At Summit, the cage effect is small (0.1 ‰) due to its high snow accumulation rate.

2) The disagreement between us and Dr. Hastings is that we don't think surface snow nitrate alone can be used to determine the degree of post-depositional processing, nor can O-isotopes be used to quantify post-depositional processing. These, however, were what Dr. Hastings's previous work relied upon to draw their

conclusions (Fibiger et al., 2013; Fibiger et al., 2016). The former is because post-depositional loss occurs not just at the surface but throughout the depth of the snow photic zone (35-45 cm at Summit), and becomes more apparent deeper in the snowpack where it is isolated from atmospheric deposition (i.e., where it experiences nitrate loss but not recycling). For the latter, O-isotopes cannot be used to quantify post-depositional processing because other factors dominate their seasonal variations. We discuss with more detail below on why O-isotopes should not be used in our point-by-point response to Dr. Hastings's specific comments.

3) Keeping in mind that this manuscript focuses on the effect of post-depositional processing on the seasonality of nitrate isotopes in snowpack, we did not ignore any relevant observations: previous publications with seasonal isotope data available were all well compared or used to constrain our model, including Hastings et al. (2004), Jarvis et al. (2009) and Geng et al. (2014). The model explains all of these observations (seasonal differences) well. What is more, the model also explained the differences in $\delta^{15}N$ of nitrate in surface snow and snow at depth observed by Jarvis et al. (2009).

Finally, we would like to also point out that, there were numerous studies at Summit that observed/measured either direct emissions of $NO_x$ and HONO from snowpack or loss of snow nitrate upon preservation (e.g., Dibb et al., 1998; Munger et al., 1999; Honrath et al., 2002; Burkhart et al., 2004; Dibb et al., 2007). Our model results are qualitatively and quantitatively consistent with these studies in terms of both the observed snowpack $NO_x$ flux and nitrate loss. In comparison, conclusions of Dr. Hastings's studies were not consistent with these observations (Hastings et al., 2004; Fibiger et al., 2013; Fibiger et al., 2016).

***Spatial heterogeneity***

*First and foremost, the issue of spatial heterogeneity in the isotopic composition of snowpits and surface snow mean that comparing with a single year of observations and suggesting that everything can be explained is not appropriate. I understand the purpose of the study here is to focus on the loss of nitrate from within the photolytic zone and the model then suggests that much of this snow-sourced NOx is re-deposited as nitrate at the surface. Hence the interest in using the surface snow observations we published in Jarvis et al. (2009). However, there are additional observations that could be compared with to better understand if the need for photolysis in the model is actually correct. For starters, Fibiger et al. (2016) present observations of d15N, d18O and D17O in surface snow and in atmospheric nitrate for two years (2010 and 2011) in the spring (May/June). Both the surface snow observations and the atmospheric observations should be compared with the model predictions. Jarvis et al (2009) also includes atmospheric observations that are neglected here.*

**Response**: There is no spatial heterogeneity of the seasonal patterns of nitrate concentrations nor isotopes at Summit. All snowpit data from Summit show a consistent

range in the **seasonality** of $\delta^{15}N$ and $\Delta^{17}O(NO_3^-)$ (Geng et al., 2014; Hastings et al., 2004; Jarvis et al., 2009; Kunasek et al., 2008). In addition, we did not use "a single year of observations", the snowpack data from Geng et al. (2014) covers 3 years, and the surface snow and snowpack data from Jarvis et al. (2009) also covers 2 years.

We didn't use the Fibiger et al. (2016) data because it does not provide any information on seasonality, which is the focus of this paper. Data from Fibiger et al. (2016) were only for May/June, and the Jarvis et al (2009) atmospheric data also only included two months (i.e., March 24 to April 21, and May 24 to July 6). Without knowing the data covering the rest of the year, or samples collected in the same season but at depth, those data are not useful for examining seasonality. Note the model meant to predict **the changes** that occur between primary deposition and preservation, it can't predict the actual values in the air **unless the values of primary nitrate are known**.

*Additionally, it is inappropriate to compare the range in surface values to the range found in snowpits and suggest that this represents real change in the isotopes (lines 65-85; ~line 240) – again there is significant spatial heterogeneity to contend with and the comparison with observations should be an envelope or a distribution and not a single line. Furthermore, it appears that one year of surface snow values is being compared to an averaged snowpit from a different year? Again, spatial AND temporal heterogeneity could easily explain a big portion of this difference (if not all). Also, the "higher" values in the snowpit than in surface snow neglect the fact that it is possible that there is also local contamination of the snow by the presence of the field camp/field work – this is well discussed and direct evidence for the isotopic values associated with this potential source are given in Fibiger et al. (2016).*

**Response**: We actually compared the annual and seasonal averages with standard deviations (±), the data were averaged with two years of surface snow data from Jarvis et al. (2009), and we did not simply compare the ranges. Regarding the spatial and/or temporal heterogeneities, **our snowpit and that reported by Jarvis et al. (2009) with which we compared were all collected in 2007 and both covered from 2004 to 2007**. Especially, the snowpit samples from Jarvis et al. (2009) and Geng et al. (2014) displayed similar $\delta^{15}N(NO_3^-)$ ranges and seasonality, and it is the Jarvis et al. snowpit data that show an enrichment in $\delta^{15}N(NO_3^-)$ when compared with their surface snow sample (Figure 5b of Jarvis et al.): among seven of eight seasons the snowpack $\delta^{15}N$ was significantly enriched compared to surface snow samples in Jarvis et al. (2009). **In the abstract of Jarvis et al. (2009), quote, "Photolytic recycling and loss will increase the $\delta^{15}N$ of buried snow nitrate, the degree to which depends on the nitrogen isotope fractionation of nitrate photolysis"**. In this study, we quantified the "degree" using the model with experimentally determined fractionation factors associated with photolysis.

Fibiger et al (2016) discussed that "local contamination of the snow by camp power generator emissions" would lead to nitrate with high $\delta^{15}N$, and Dr. Hastings suggested here this could possibly explain the high snowpit $\delta^{15}N$ values. However, contamination is unlikely to explain the seasonality given that several snowpits collected and measured by three different groups show similar seasonality and high $\delta^{15}N$. In addition, among

seven of eight seasons the snowpack $\delta^{15}N$ was significantly enriched compared to surface snow samples in Jarvis et al. (2009). This is impossible to explain with contamination.

**Fpri assumptions**

*The assumptions of the values for Fpri (both concentration and isotopically) need to be better justified. The Fibiger et al. (2016) surface snow isotope data ALL fall within a 3-end member plot (see Figure 8 in that paper) that are suggested to represent the primary nitrate signatures (this includes D17O data from Fibiger et al. (GRL, 2013) as well). Why not use these end-members as an a priori assumption/test? Why assume surface snow values when authors of the current work have argued in other papers that even the top few cm misrepresents what is happening in the very surface layer of the snow? It is stated in this paper that the Fpri assumptions could be underestimates – which directly agrees with the values presented in Fibiger et al. So again, why not use these values compared to the constant 0 and 30 per mil assumptions for d15N and D17O, respectively. In fact, if the highest endmember values for D17O found in Fibiger et al (2016) (39 per mil) were used for D17O primary it likely would much better fit the seasonal D17O observations than the model currently does! This offers much in the way of better constraining our understanding of nitrate in snow at Summit that weakens the current manuscripts as presented.*

**Response**: The Fibiger et al. (2016) reported measurement of nitrate isotopes ($\delta^{15}N/\Delta^{17}O$) in atmosphere and surface snow at Summit over two months (May to June), making it impossible to use this dataset to inform the seasonal variations in isotopes of primary nitrate. Instead, the Jarvis et al (2009) gave seasonal isotope values from surface snow that we used as sensitivity tests, and we have stated clearly that the surface snow values do not represent that of primary nitrate, but represent the intermediate state between primary nitrate and the final preserved nitrate.

Regarding $\Delta^{17}O(NO_3^-)$, as we have stated in the manuscript and earlier in our response, the purpose of this manuscript was to investigate the effect of post-depositional processing on the seasonality of snowpack nitrate and its isotopes, and the TRANSITS model applied to this study was never aimed to predict the isotopes of snow nitrate, but to assess the **changes** caused by post-depositional processing upon the preservation of nitrate. What is more, it is well understood that photo-driven post-depositional processing can lead to large N isotope fractionation, while not directly affecting $\Delta^{17}O$ that is a mass-independent fractionation signal (Berhanu et al., 2015). Therefore, we should expect a large isotope affect from post-depositional processing on $\delta^{15}N$ but not on $\Delta^{17}O$, which is what exactly the model predicted.

**D17O mismatch**

*A critical point is that the model cannot match the D17O observations. The authors note that a primary signal therefore must be important in driving the seasonality. Yet, the global chemical modeling results of Alexander et al. (2020) do not match the D17O observations at Summit either, and GEOS-Chem does not include exchange with the snow/post-depositional processing of nitrate. Why not discuss these results compared*

*to the understanding trying to be made locally here? With neither a chemical transport model nor a box model matching the results our understanding clearly has a long way to go and there is an important opportunity for the authors to make an advance here. For instance, the key point in the beginning is that post-depositional processing does not significantly affect D17O. What is significant? The GEOS-Chem model simulations predict the seasonality of D17O well but overestimates the values; would the 2.1 per mil decrease here because of recycling help to resolve much of this over-estimation? If it does, isn't that significant? The lack of significance seems to be related to the fact that 2.1 per mil is much different than the 9 per mil difference seen in the observations, but the co-authors of this work have used changes as small as 2 per mil to argue for significant changes in atmospheric oxidation pathways in other environments. In fact, in the spring surface snow alone the median D17O values between 2010 and 2011 change by ~3 per mil (Fibiger et al., 2013) – which again can reflect spatial and temporal heterogeneity OR differences in chemistry between years (both of which are not considered in this study).*

**Response**: Please see our response to the last comment. Again, it is not our objective to predict the snow $\Delta^{17}$O, but instead to predict **the changes induced by post depositional processing**. What Dr. Hastings asked would be best addressed through incorporation of the snow column model into a chemical transport model such as GEOS-Chem, which is out of the scope of this manuscript.

In addition, "significance" used here is to represent a relative term, and "significant or not" depends on what it is compared with. For averages over a period of hundreds of years, 2 permil difference is significant; while for a seasonal difference of > 9 permil caused mainly by seasonal formation pathway differences, 2 permil difference caused by the post-depositional processing is less significant. Since significance is a statistical term and here we don't use is statistical sense, we will replace "significant" with "important" in the revised manuscript such as "post-depositional processing is not **important** in determining the $\Delta^{17}$O seasonality".

**Other fractionation effects**

*The authors discuss re-formation of nitrate in the snow – does this induce an isotopic effect on d15N? It should be addressed that re-formation within the snow that impacts D17O should also impact d15N (and d18O) as well.*

*The potential for loss via evaporation/volatilization of nitrate should also be considered (Shi et al., Isotope fractionation of nitrate during volatilization in snow: A field investigation in Antarctica. Geophys Res Lett. 46(6):3287-3297. 10.1029/2019GL081968, 2019)*

**Response**: There are essential differences in nitrogen and oxygen during reformation of snow nitrate. The recombination effect of snow nitrate would induce change in $\Delta^{17}$O because of the secondary chemistry during reformation of nitrate (McCabe et al., 2005; Meusinger et al., 2014), such as exchange with oxygen isotopes of ice. But from the Meusinger et al. experiment, there is no evidence that nitrogen isotopes are altered during reformation. If the reformation is quantitative, one would not expect an isotope effect.

We have added some discussions for physical loss of snow nitrate in our revised manuscript although this study focuses on photolysis impact on snow nitrate, which is thought to be the dominant post depositional process (Erbland et al., 2013; Frey et al., 2009; Berhanu et al., 2015). The Shi et al. (2019) experiment was flawed. They collected snow at the inland Dome A location and put it in an open-door room at Zhongshan Station (a coastal site). Dome A snow is of much higher nitrate concentrations than in snow at the coast (500 ppb at Dome A while 30 ppb at the coast, Shi et al., 2015); thus, a re-equilibrium between snow and the overlying air will be established which will artificially enhance the physical release (evaporation).

In addition, in both evaporation experiments of Shi et al. 2019 and earlier by Erbland et al. (2013), when mass loss is significant at higher temperature, the fractionation factor is very small. At Summit, even using a 25 % mass loss from evaporation (the maximum loss fraction given by Dibb et al. (2007)) and a fractionation constant of 3.6 ‰ (Erbland et al., 2013), only a 1 per mil difference in $\delta^{15}N$ can be induced, which is negligible compared to the photolysis induced fractionation.

**The conclusion of 21% loss**

*The key conclusion of the manuscript is that "as much as 21% of nitrate is lost". In the model the majority of this nitrate is not actually lost – it's redistributed. And the model better explains the seasonality based on a primary signal change. On net, more like **2% of nitrate is actually lost and this agrees** with what Fibiger et al (2016)'s observational study based on isotopes found and what Thomas et al. (2011)'s modeling study based on gas phase and snow concentrations of a variety of species found.*

**Response**: We stated clearly in abstract and conclusion that 'up to 21 % of nitrate was **lost and/or redistributed** after deposition', and this ratio of loss is referred to a specific (seasonal, monthly or finer resolution) layer below the photic ozone. This value is consistent, or at least within the range of nitrate loss estimated by Dibb et al. 2007 who compared surface snow nitrate concentrations with concentrations at depth. For **annual net loss** it is 4.1 %, assuming 35 % snow sourced nitrate was transported away (export). The fraction of export could be larger, as suggested by Honrath et al. (2002) who concluded that without wet deposition such as snowfall or fog deposition the $NO_x$ or $HNO_3$ emitted from sunlit snow should be largely exported from local boundary layer at Summit.

When focusing on the seasonality of the isotopes in our manuscript, it is important to distinguish net annual loss from loss of a specific layer. The **net** annual loss can be small since recycling occurs, but the nitrate at depth gets redistributed to the surface snow. Therefore, for specific snow layers below the surface, the loss is larger and so are the isotope effects. It is the loss of a specific layer that determines the **changes** in $\delta^{15}N$ from nitrate deposition to its final preservation and influences $\delta^{15}N$ seasonality. This is the entire point of this manuscript. Given the seasonally varying actinic flux and snow accumulation rate, we think our calculation is reliable and agrees well with previous studies.

In addition, we would like to point out that the conclusion of '*2% of nitrate is*

***actually lost'*** from Fibiger et al. (2013) was likely flawed. Fibiger et al. (2013) estimated the loss fraction by multiplying 0.1 % (loss fraction in 3 days in the upper 10 cm snow, they cited from Thomas et al. (2011)) with a factor of 21 (resident time in photic zone at Summit). The '0.1 %' they used in this calculation is a severe underestimate. From the supplemental file of Thomas et al. (2011) (Figure 9), the '0.1 %' value actually should be around 1% to 2%.

*Quoting here from the discussion section of Fibiger et al (2016): "Previously, it was thought that local recycling of NO3 might be important at Summit [Jarvis et al., 2009; Kunasek et al., 2008]. If this were true, however, there should be some connection between local gas phase concentrations and the isotopes of NO3 in the snow. If HNO3 were formed locally and deposited by cloud-to-ground scavenging of NO3 in the snow (Figure 1, arrows d and g), then BrO concentrations above 1 pptv should be influencing NO3 in the snow [Kunasek et al., 2008; Morin et al., 2007] via reactions (R6) through (R9). In particular, we expect that when BrO is high, the Δ17O and δ18O of nitrate would also be high, as BrO retains the anomalous isotopic signature of the O3 from which it is derived. The local signal, if important, should be present in the snow as the lifetimes of NO and HNO3 at Summit are only a few hours. This is evident in the atmospheric HNO3 and NO concentrations at Summit, as both approach zero at low solar zenith angle. This is evidence that there is some loss or recycling of NO3 from the snow in Greenland [Honrath et al., 1999], but as noted above, as little as 2% of NO3 loss from the snow can account for observed NOx concentrations above the snow [Thomas et al., 2011]. This photolysis of NO3 to NOx has a significant influence on local NOx concentrations and the δ15N-HNO3 in the atmosphere at Summit, but appears small enough to not have a significant effect on the residual NO3 in the snow. If photolysis of NO3 to NOx followed by deposition of locally formed HNO3 (Figure 1, arrows a, c, and d) was having a strong influence on the NO3 in the snow, we would expect that snow NO3 concentrations would reflect NO and HNO3 atmospheric concentrations. There was, however, no connection found between the local concentrations of BrO, NO, or NOy and any of the isotopes of NO3 or [NO3]. This lack of relationship was found using 3, 5, and 12 h back averages of the gas phase data, from each time point that a snow sample was taken, accounting for potential variations in the lifetime of NOx against deposition as NO3. This indicates that local chemistry, either through recycling of NO3 or local conversion of NOx to NO3, is not influencing the NO3 preserved in the snow. This lack of relationship is true both across each season and over shorter timescales within." In the end, our work has suggested that photolytic loss and recycling may be taking place but it represent a very small portion of the overall nitrate in the snow. The authors of present student should be obliged to provide clear evidence as to why this conclusion is not justified.*

**Response**: As responded earlier, the Fibiger et al (2013, 2016) studies were flawed by the fact that only atmospheric and surface snow samples were used and assessed, while it is the samples below the photic zone that should be used to fully quantify the effect of post-depositional processing. In addition, the arguments themselves used by

these studies were also with faults that we describe in details as follows:

Both Fibiger et al. (2013) and Fibiger et al. (2016) concluded that post-depositional processing was negligible at Summit, and the arguments were mainly based on O-isotopes. Fibiger et al. (2013) drew their conclusion because they observed a strong relationship between $\Delta^{17}O$ and $\delta^{18}O(NO_3^-)$ in their **surface snow samples.** But we don't agree that this is evidence of small to no photolysis. First of all, in East Antarctica where severe post-depositional processing occurs, strong correlations between $\Delta^{17}O$ and $\delta^{18}O(NO_3^-)$ in snow and/or aerosol were also observed (Erbland et al., 2013; Frey et al., 2009). **In particular, similar correlations between $\Delta^{17}O$ and $\delta^{18}O(NO_3^-)$ were found in atmospheric and surface snow nitrate at Dome C (attached Figure 1.1, 1.2 and 1.3), where photolysis of snow nitrate has been unambiguously proved to be dominant** (Erbland et al., 2013). So correlations between $\Delta^{17}O$ and $\delta^{18}O(NO_3^-)$ should be viewed as direct transfer of atmospheric signal to surface snow and cannot be viewed as evidence of little to no post-depositional processing.

Second, at Summit, the strong correlation between $\Delta^{17}O$ and $\delta^{18}O$ of nitrate only presents in surface snow ($R^2 = 0.9$) and becomes weaker ($R^2 = 0.32$) at depth (Attached Figure 2), suggesting that post-depositional processing takes effect over the duration of the residence time of nitrate in the snow photic zone. This is consistent with the observations at Dome C where the correlations get worse from atmospheric nitrate to snowpit nitrate (Figure 1.1., 1.2 and 1.3). Overall, given the snow accumulation rate at Summit, nitrate stays at the surface (1~2 cm) by only ~one week on average, while it takes nitrate about half year to be buried beneath the photic zone and be ultimately preserved.

Regarding the Fibiger et al. (2016) study, they concluded there is little to no local recycling snow nitrate because that they did not observe increases in snow $\Delta^{17}O(NO_3^-)$ when **atmospheric** BrO concentration increased by ~ a few ppt. However, such argument is biased for several reasons and their assessment was not quantitative. First, BrO originates from local photochemistry as it has a very short lifetime and can't be from long-range transport, and the production of BrO will consume $O_3$. This is a tradeoff regarding their effects on $\Delta^{17}O$ of $NO_2$ which determines $\Delta^{17}O(NO_3^-)$. Indeed, BrO could also form nitrate through hydrolysis of $BrONO_2$, which will increase $\Delta^{17}O(NO_3^-)$. But when BrO concentration increased, how did OH and $HO_2/RO_2$ change as they all related to photochemistry? Observations at Summit suggested BrO concentration always co-varied with OH/ $HO_2/RO_2$ (Liao et al., 2011) because they were both controlled by local actinic flux conditions. If OH and $HO_2/RO_2$ concentration also increased (the authors didn't assess these radicals) at the same time, they would decrease $\Delta^{17}O(NO_3^-)$. Under this circumstance, increases in snow $\Delta^{17}O(NO_3^-)$ is unlikely to occur.

Second, whether or not the reformed nitrate in the air during the short duration of increased BrO (only a few hours) was able to influence local nitrate budget is unknown. Our calculation suggested that the locally formed nitrate can account for at most 25% of the deposited nitrate in summer and the rest is from transport. Additional nitrate formed from the increased BrONO2 hydrolysis is only a part of this locally formed

nitrate, and whether its effects on $\Delta^{17}O$ is detectable or not has to be carefully assessed, which is beyond the scope of the paper.

Third, the $\Delta^{17}O(NO_3^-)$ of surface snow (1-3 cm) was used by Fibiger et al. (2016) to compare with the effect of atmospheric BrO concentration increase. Honrath et al. (2002) reported the daily dry depositional flux of atmospheric nitrate at Summit to be $7.16 \times 10^{11}$ molecules $m^{-2}$ $s^{-1}$. If **no wet deposition** occurs, the **newly deposited nitrate** during their sampling period (4 to 12 hours) only constitutes a minor fraction of surface snow nitrate: average surface snow (top 2 cm) nitrate concentration at Summit is 100 ppb, which takes about 94 days for dry deposition of atmospheric nitrate to accumulate to this amount. Therefore, even if atmospheric nitrate $\Delta^{17}O$ indeed increased because of the BrO concentration increase over a duration of a few hours, it is unlikely to be detectable in snow sample.

Therefore, the conclusion of Fibiger et al. (2016) study was not reliable either. In addition, quoting here the Jarvis et al. 2009 paper that Dr. Hastings is the second author, in their abstract: "*Measurements of isotopically labeled nitrate in surface snow confirm that photolytic recycling of snowpack nitrate, in which photolyzed nitrate products recombine to form HNO₃ that is subsequently redeposited to the snow surface, does occur at Summit*". There is a much larger N-isotope effect associated with the post-depositional processing compared to O-isotopes, and it is also the nitrogen isotopes that should be used to reflect and/or assess the effect of post-depositional processing as has been done in Antarctica.

In summary, neither Fibiger et al. 2013 nor 2016 studies can really inform the actual degree of post-depositional processing at Summit for the reasons stated above.

[Figure]

**Figure 1.1** Scatter plot of $\delta^{18}O(NO_3^-)$ versus $\Delta^{17}O(NO_3^-)$ in the atmosphere at Dome C, Antarctic (Erbland et al., 2013).

[Figure]

**Figure 1.2** Scatter plot of $\delta^{18}O(NO_3^-)$ versus $\Delta^{17}O(NO_3^-)$ in the surface snow at DC. Note the relatively low snow accumulation rate at DC (~ 10 cm per year) leads to significant post-depositional processing thereby. However, the strong correlation between $\delta^{18}O(NO_3^-)$ and $\Delta^{17}O(NO_3^-)$ similar to Summit is also observed, suggesting the uppermost snow nitrate doesn't undergo much post-depositional processing but a direct representation of atmospheric nitrate signals.

[Figure]

**Figure 1.3** Scatter plot of $\delta^{18}O(NO_3^-)$ versus $\Delta^{17}O(NO_3^-)$ in the bulk snowpack at DC. The strong correlation weakens when referring to the whole snowpack as a result of post-depositional processing (Erbland et al., 2013).

[Figure]

**Figure 2**. Correlation between $\delta^{18}O(NO_3^-)$ versus $\Delta^{17}O(NO_3^-)$ in the bulk snowpack (~ 2m) at Summit (Geng et al., 2014). This correlation is much weaker than that in surface snow reported by *Fibiger et al.* 2013 ($r^2$=0.9). The $\Delta^{17}O$ and $\delta^{18}O(NO_3^-)$ correlation at depth becomes worse, and which reflects the effect of post-depositional processing.

*An approach that truly considers past work and challenges/reconsiders previous interpretations, based on a robust comparison between the model and observations, is needed and not provided in this manuscript. Rather, model results are presented that 1) show that the isotopic composition of preserved nitrate is quite sensitive to the assumed signature in nitrate when first deposited, and 2) suggest that photolysis of nitrate may significantly modify the seasonal profile in d15N if enough of the nitrate in the snow is photolyzed.   The authors assert that the seasonal variation in the isotopic composition of deposited nitrate is too poorly constrained to consider that it might be preserved largely unaltered (our conclusion from multiple prior studies), and conclude that the observed seasonal pattern is created by post-depositional photolysis (their assumption at the outset, neglecting prior peer-reviewed work).*

 **Response**: We appreciate the large body of studies conducted by Dr. Hastings's group, but again the "*conclusion from multiple prior studies*" were mainly based on surface snow samples and O isotopes that are not suitable to determine the effect of post-depositional processing, and can't be simply compared with the TRANSITS model results. We have argued this in the introduction part of our manuscript.

 The seasonality in $\delta^{15}N$ on which the model focused is independent of the values of primary nitrate $\delta^{15}N$, so is the difference in $\delta^{15}N$ between primary nitrate and the finally preserved nitrate (i.e., PIE, the photo-induced isotope effect in the model). The modeled photolysis of snow nitrate is well constrained by, and comparable with the observed snowpack NOx emissions, and the modeled seasonality and $\delta^{15}N$ change upon preservation compare well with observed snowpack $\delta^{15}N$ seasonality (Hastings et al., 2004, Jarvis 2009 and Geng et al. 2014), and the observed difference in $\delta^{15}N$ between

surface snow nitrate and that at depth (Jarvis et al., 2009), respectively. Therefore, we think we present a robust comparison between the model and observations, and did not ignore any prior peer-reviewed work. The Fibiger et al (2013, 2016) studies we have briefly discussed in the introduction and stated why we don't agree with their conclusions. We did not compare the data in these two studies though, because they only reported data in a couple of months and no seasonal information can be informed, and thus are not suitable to include in the model comparison.

**Reference**

Berhanu, T. A., Savarino, J., Erbland, J., Vicars, W. C., Preunkert, S., Martins, J. F., & Johnson, M. S. (2015). Isotopic effects of nitrate photochemistry in snow: a field study at Dome C, Antarctica. *Atmospheric chemistry and physics*, *15*(19), 11243-11256.

Burkhart, J. F., M. Hutterli, R. C. Bales, and J. R. McConnell (2004), Seasonal accumulation timing and preservation of nitrate in firn at Summit, Greenland, *Journal of Geophysical Research-Atmospheres*, *109*(D19), D19302.

Chu, L., & Anastasio, C. (2003). Quantum yields of hydroxyl radical and nitrogen dioxide from the photolysis of nitrate on ice. The Journal of Physical Chemistry A, 107(45), 9594-9602.

Dibb, J. E., R. W. Talbot, J. W. Munger, D. J. Jacob, and S. M. Fan (1998), Air-snow exchange of $HNO_3$ and $NO_y$ at Summit, Greenland, *Journal of Geophysical Research-Atmospheres*, *103*(D3), 3475-3486.

Dibb, J. E., et al. (2007), An overview of air-snow exchange at Summit, Greenland: Recent experiments and findings, *Atmos. Environ.*, *41*(24), 4995-5006.

Fibiger, D. L., M. G. Hastings, J. E. Dibb, and L. G. Huey (2013), The preservation of atmospheric nitrate in snow at Summit, Greenland, *Geophys. Res. Lett.*, *40*(13), 3484-3489.

Fibiger, D. L., J. E. Dibb, D. X. Chen, J. L. Thomas, J. F. Burkhart, L. G. Huey, and M. G. Hastings (2016), Analysis of nitrate in the snow and atmosphere at Summit, Greenland: Chemistry and transport, *Journal of Geophysical Research-Atmospheres*, *121*(9), 5010-5030.

Hastings, M. G., E. J. Steig, and D. M. Sigman (2004), Seasonal variations in N and O isotopes of nitrate in snow at Summit, Greenland: Implications for the study of nitrate in snow and ice cores, *J. Geophys. Res.*, *109*(D20), D20306.

Honrath, R. E., Y. Lu, M. C. Peterson, J. E. Dibb, M. A. Arsenault, N. J. Cullen, and K. Steffen (2002), Vertical fluxes of NOx, HONO, and HNO3 above the snowpack at Summit, Greenland, *Atmos. Environ.*, *36*(15-16), 2629-2640.

Jarvis, J. C., M. G. Hastings, E. J. Steig, and S. A. Kunasek (2009), Isotopic ratios in gas-phase HNO3 and snow nitrate at Summit, Greenland, *Journal of Geophysical Research: Atmospheres*, *114*(D17), D17301.

Liao, J., Huey, L. G., Tanner, D. J., Brough, N., Brooks, S., Dibb, J. E., ... & Gorham, K. (2011). Observations of hydroxyl and peroxy radicals and the impact of BrO at Summit, Greenland in 2007 and 2008. *Atmospheric Chemistry and Physics*, *11*(16), 8577-8591.

Meusinger, C., Berhanu, T. A., Erbland, J., Savarino, J., & Johnson, M. S. (2014). Laboratory study of nitrate photolysis in Antarctic snow. I. Observed quantum yield, domain of photolysis, and secondary chemistry. The Journal of chemical physics, 140(24), 244305.

Munger, J. W., D. J. Jacob, S. M. Fan, A. S. Colman, and J. E. Dibb (1999), Concentrations and snow-atmosphere fluxes of reactive nitrogen at Summit, Greenland, *Journal of Geophysical Research: Atmospheres*, *104*(D11), 13721-13734.

---

## Author Comment (AC4)

Dear Dr. Dominé,

We appreciate your recommendation, and are grateful to the two anonymous referees and Dr. Hastings for their comments.

We have made point to point responses to the two anonymous referees, and think we have addressed their concerns by adding more details on how the wavelength-dependent e-folding depth was computed in the model, as well as a summary table presenting all necessary parameters used to constrain the model, etc. Details can be found in our point-to-point response to these referees.

Regarding to Dr. Hastings's comments, apparently there is a strong disagreement between the two groups (us vs. Dr. Hastings), and we appreciate the questions raised here as it offers an opportunity to discuss with details. First, we thought Dr. Hastings may have misunderstood the purpose of this manuscript as well as the TRANSITs model that we used. The manuscript and the model were aiming to predict the **changes** of isotopes of nitrate from its primary deposition to final preservation, and to what degree the magnitude of the observed seasonality can be explained by post-depositional processing. Keeping this in mind, we did not ignore any relevant observation and the model agreed with observations just well. Second, we don't agree with Dr. Hastings' previous conclusion that there is little to no changes caused by post-depositional processing. Because neither surface snow nitrate alone can be used to determine the degree of post-depositional processing, nor can O-isotopes be used to quantify post-depositional processing. These, however, were what Dr. Hastings's previous work relied upon to draw their conclusions (Fibiger et al., 2013; Fibiger et al., 2016). We have made specific rationales in our point-to-point response to Dr. Hastings to elucidate the above-mentioned points.

We are looking forward to hearing from you, and any additional suggestions/comments improving this manuscript would be highly appreciated.

Best regards,
Authors

---

## Author Response (AR2)

We are grateful to the editor's comments. Below we list detailed responses to the editors' comments. The comments are listed in italics, followed by the response in normal font with changes highlighted in blue.

**Response to Editor's comments**
**Comments**

*Thank you for this extensively revised version. The main issue was in fact the comments by Dr. Hastings. While there is clearly room for further discussion, I believe that you have adequately addressed the reviewers' comments as well as Dr. Hastings' ones. I think your work represents a valuable contribution to this active research field. You have explained the scopes and limits of your model with more clarity and also given more detail on Dr. Hastings' work. This will probably help the reader forge his/her own opinion on the respective merit of each group's contributions. I am therefore pleased to accept your paper for publication in The Cryosphere, after you have addressed the minor points below. One of these points is on your calculation of snow specific surface area (SSA). In a way, I perhaps should have pointed that out sooner. However, I was in a position of conflict of interest, since your SSA calculations involves my own work, and Editors are in principle not supposed to promote their own work when editing papers, for obvious ethical reasons. However, if you agree that my comment is in the interest of the quality of your paper, I let you think about it. Obviously if you do so, you will cite (Carmagnola et al., 2013), of which I am the second author, and if you think this is self-serving, please just skip this comment.*

**Response**: Thanks for the comments. The Carmagnola et al. (2013) has been cited in our previous version manuscript for the reported dust concentration within. We agree that the measured SSA at Summit in this work provides extra constrains for calculating the radiative transfer in our model and we are pleasant to add some discussions about it. Please see our detailed response below.

**Comments**

*Line 52: why an error margin for d15N and not for d18O?*

**Response**: This is because for $\delta^{15}N$ the value used here is adopted from the laboratory experiment results under Dome C conditions with error margin (Berhanu et al., 2014). While Berhanu et al. (2014) observed clear isotope fractionation for $\delta^{15}N$ during the photolysis experiments, they didn't find systematic change in $\delta^{18}O$, so we used the theoretical photolysis fractionation factor for $\delta^{18}O$ from Frey et al. (2009) as the representative, which was derived with DC actinic flux conditions and different cross sections for $^{18}O$-substituted $NO_3^-$ isotopologues and no error margin reported.

**Comments**

*Line 178. I just realized that your SSA values used to calculate your e-folding depth for actinic flux used equation (3) of (Domine et al., 2007). However, that equation may not be the most appropriate. First, the work of (Domine et al., 2007) is for seasonal snow, which may be different from snow at Summit. Second, you use equation (3) proposed for fresh snow, while there is not much fresh snow over the snow depth you consider. A better approach would have been to consider snow type as well as snow density, and use the most adequate equation. In any case, given the snow types at Summit, equations (4), (5) or (8) would probably have been more appropriate. These would have yielded lower SSA values and a greater e-folding depth. Then, since these equations may in fact not apply perfectly to Summit, it would be possible to compare*

*the SSA values obtained with those found by (Carmagnola et al., 2013) at Summit in spring (see their Figure 8). If their density values are similar to yours, then their SSA values would probably be a better estimate than using equation (3) of Domine. Also please mention that that equation has units of cm2 kg-1 for SSA and g cm-3 for density. Your stated SSA values of 44 to 51 m2 kg-1 in any case appear quite high for Summit, as (Carmagnola et al., 2013) found most photic zone values in the range 20 to 40 m2 kg-1. At this stage, it may not be appropriate to change your SSA values, but you may at least want to briefly evaluate the impact of using lower SSA on the e-folding depth and on your overall results. You may conclude that measuring SSA values would be desirable, but that currently the lack of measurements and uncertainties in estimates may modulate your results by a given amount. Depending on the results, you may add or not SSA as a source of uncertainty in your conclusions, e.g. line 452.*

**Response**: we appreciate the editor for pointing out the limitation in our calculation of snowpack SSA that used in modelling radiative transfer in snow. There are several regression relationships between seasonal snowpack density and SSA summarized by Domine et al. (2007) according to snow groups, namely fresh snow (type F), recent snow (type R) and aged snow (type A). The reason that we have chosen equation (3) to get SSA according to our measured snow density profile is because it yielded an e-folding depth of 12.3 cm at 305 nm wavelength that is close to the measured values ($11.6 \pm 2.6$ cm) by Galbavy et al. (2007), despite the predicted SSA was higher than the values reported by Carmagnola et al. (2013). In comparison when using either equitation (5) or (8) that are for aged snow which yielded SSA values (4-20 $m^2$ $kg^{-1}$) that are significantly lower than the values reported by Carmagnola et al. (2013), and the resulting e-folding depth (i.e., 20.6 cm and 36.0 cm, respectively) are too high compared to the measured values (attached figure 1). While using equation 4 yield SSA values of 19 to 27 $m^2$ $kg^{-1}$ and the corresponding e-folding depth of 18.0 cm, still much higher than the observed values by Galbavy et al. (2007).

The mean snow density reported by Carmagnola et al. (2013) was 330 kg $m^{-3}$ in the top 50 cm, much lower than our snowpack (395 kg $m^{-3}$). In addition, the black carbon content measured in Carmagnola et al. (2013) (0.3 ng $g^{-1}$ in average) is also relatively low comparing to the value used in this study (1.4 ng $g^{-1}$, according to Zatko et al., 2013). We acknowledge all these differences would introduce differences in the modeled e-folding depth, and it is probably the choosing of Eq(3) compensated other uncertainties, resulting in an e-folding depth that best matches the observations.

In order to remind the readers the uncertainties regarding the calculation of snow radiative transfer, in the revised manuscript we have made changes as follows:

Line 172: "we used the regression relationship between SSA and $\rho_{snow}$ (SSA = $-174.13 \times$ ln($\rho_{snow}$) + 306.4, in unit of cm$^2$ g$^{-1}$ for SSA and g cm$^{-3}$ for density, respectively) from Domine et al. (2007) to calculate SSA."

Line 179-187: "This likely explains why our calculated e-folding depth was smaller than Zatko et al. (2013) despite using the same impurity content. Note the regression relationship between SSA and $\rho_{snow}$ from Domine et al. (2007) was for fresh snow, which may not be suitable for SSA prediction for the whole snowpack. However, using this equation yielded an e-folding depth that is similar to the observations by Galbavy et al. (2007), despite the yielded SSA appears to be larger than the observed values (20 to 40 $m^2$ $kg^{-1}$) by Carmagnola et al. (2013) for a Summit

snowpack which has a much lower snow density (averaged 330 kg m$^{-3}$ in the top 50 cm) than ours (averaged 395 kg m$^{-3}$). Nevertheless, given the uncertainties related to the calculation of snow radiative transfer that are currently not well constrained, the regression relationship between SSA and $\rho_{snow}$ used here yielded a reasonable e-folding depth similar to the observations. Improvements can be made if snow physicochemical properties (e.g., SSA, density, and impurities concentrations) can be precisely well constrained by future observations."

Line 451: "Despite uncertainties in the model, e.g., specific surface area, quantum yield of snow nitrate photolysis, the export fraction, the modeled loss/redistribution of nitrate after deposition is consistent with previous studies, and explains the observed difference between $\delta^{15}N(NO_3^-)$ in surface snow and snow at depth."

Line 471-473: "On the other hand, precise measurements of snowpack properties, e.g., specific surface area, impurity concentrations, observational constraints on the quantum yield of snow nitrate photolysis, and better constraints on the export fraction are also needed in order to improve the model's performance."

[Figure]

**Figure1**. Top: measured snowpack density profiles in Geng et al. (2014) and the calculated SSA according to the SSA-density regression relationships from Domine et al. (2007). Black: density; other lines indicate SSA using different regression relationships. Bottom: modelled snow actinic flux profile at 305 nm using different SSA-density regression relationships. The calculated e-folding depth is shown in the legend. Type F: fresh snow; type R: recent snow; type A: aged snow (Domine et al., 2007).

**Comments**

*Line 464. "Overall, the model results suggest an important (if not dominant) role of post-depositional processing". By "if not dominant", do you mean "although not dominant" or "perhaps even dominant"? Please clarify.*

**Response**: We meant "perhaps even dominant" here because the modeled magnitude of seasonal $\delta^{15}N(NO_3^-)$ difference is ~17.5 ‰ (without involving changes in $\delta^{15}N$ of $F_{pri}$) that is similar to the observations (16.1 ± 3.6) ‰ seasonality. We change the phrase in line 462 as follows:

"Overall, the model results suggest an important, perhaps even dominant role of postdepositional processing in regulating the snowpack $\delta^{15}N(NO_3^-)$ seasonality at Summit".

**Reference**

Berhanu, T. A., Meusinger, C., Erbland, J., Jost, R., Bhattacharya, S., Johnson, M. S., & Savarino, J.: Laboratory study of nitrate photolysis in Antarctic snow. II. Isotopic effects and wavelength dependence, J. Chem. Phys., 140, 244306, https://doi.org/10.1063/1.4882899, 2014.

Carmagnola, C., Domine, F., Dumont, M., Wright, P., Strellis, B., et al.: Snow spectral albedo at Summit, Greenland: measurements and numerical simulations based on physical and chemical properties of the snowpack, The Cryosphere., 7, 1139-1160, https://doi.org/10.5194/tc-7-1139-2013, 2013.

Domine, F., Taillandier, A. S., & Simpson, W. R.: A parameterization of the specific surface area of seasonal snow for field use and for models of snowpack evolution, J. Geophys. Res. Earth Surf., 112, F02031, https://doi.org/10.1029/2006JF000512, 2007.

Frey, M. M., Savarino, J., Morin, S., Erbland, J., & Martins, J.: Photolysis imprint in the nitrate stable isotope signal in snow and atmosphere of East Antarctica and implications for reactive nitrogen cycling, Atmos. Chem. Phys., 9, 8681-8696, https://doi.org/10.5194/acp-9-8681-2009, 2009.

Galbavy, E. S., Anastasio, C., Lefer, B. L., & Hall, S. R.: Light penetration in the snowpack at Summit, Greenland: Part 1: Nitrite and hydrogen peroxide photolysis, Atmos. Environ, 41, 5077-5090, https://doi.org/10.1016/j.atmosenv.2006.04.072, 2007.

Geng, L., Cole-Dai, J., Alexander, B., Erbland, J., Savarino, J., et al.: On the origin of the occasional spring nitrate peak in Greenland snow, Atmos. Chem. Phys., 14, 13361-13376, https://doi.org/10.5194/acp-14-13361-2014, 2014.

Zatko, M. C., Grenfell, T. C., Alexander, B., Doherty, S. J., Thomas, J. L., & Yang, X.: The influence of snow grain size and impurities on the vertical profiles of actinic flux and associated $NO_x$ emissions on the Antarctic and Greenland ice sheets, Atmos. Chem. Phys., 13, 3547-3567, https://doi.org/10.5194/acp-13-3547-2013, 2013.